# LEARNING THE GREATEST COMMON DIVISOR: EXPLAINING TRANSFORMER PREDICTIONS

**François Charton**
Meta AI
`fcharton@meta.com`

## ABSTRACT

The predictions of small transformers, trained to calculate the greatest common divisor (GCD) of two positive integers, can be fully characterized by looking at model inputs and outputs. As training proceeds, the model learns a list $\mathcal{D}$ of integers, products of divisors of the base used to represent integers and small primes, and predicts the largest element of $\mathcal{D}$ that divides both inputs. Training distributions impact performance. Models trained from uniform operands only learn a handful of GCD (up to 38 GCD $\leq$ 100). Log-uniform operands boost performance to 73 GCD $\leq$ 100, and a log-uniform distribution of outcomes (i.e. GCD) to 91. However, training from uniform (balanced) GCD breaks explainability.

## 1 INTRODUCTION

Transformers (Vaswani et al., 2017) have been applied to problems of mathematics, both symbolic (Lample & Charton, 2019; Charton et al., 2020; Shi et al., 2021) and numerical (Charton, 2021). Yet, they struggle with basic arithmetic (Lee et al., 2023; Nogueira et al., 2021). Large language models (LLM) can learn addition or multiplication by a small prefactor, and generalize beyond their training range when fine-tuned using scratchpad (Nye et al., 2021), chain-of-thought (Wei et al., 2023) or algorithmic prompting (Zhou et al., 2022), but these techniques require bespoke data and do not extend to complex tasks (Dziri et al., 2023). Math transformers were also found to be brittle (Welleck et al., 2021), to fail on simple tasks (Davis, 2023), and to be hard to interpret, except in the simplest cases (Nanda et al., 2023). Yet, small transformers can learn advanced calculations, such as eigen-decomposition (Charton, 2021) and polynomial roots (Charton, 2022b).

In this paper, I train 4-layer transformers to compute the greatest common divisor (GCD) of two positive integers, an important operation for rational arithmetic and number theory, and observe that:

1. **Transformers learn to cluster input pairs with the same GCD.** All pairs of integers $(a, b)$ with the same GCD $k$ are predicted the same.
2. **Transformer predictions can be fully characterized.** During training, the model learns a set of integers $\mathcal{D}$, and predicts, for any input pair $(a, b)$, the largest element in $\mathcal{D}$ that divides $a$ and $b$.
3. Early during training, **transformers learn to predict products of divisors of the base used to represent integers**. **Small primes are "grokked"** (Power et al., 2022) after extended training.
4. **Models trained from log-uniform operands and outcomes achieve better performance.** They correctly predict up to 91 GCD $\leq$ 100. Model predictions remain fully explainable.
5. **An unbalanced distribution of outcomes in the training set is required for full explainability:** explainability partially fails once models are trained from uniformly distributed GCD.

These results demonstrate how transformers can be trained to perform exact calculations involving integer divisibility, a central task in integer arithmetic and number theory. Beyond GCD calculations, the broader potential impact of this research extends in three directions. First, it presents a new approach to model explainability: fully characterizing black-box model predictions by experimenting with selected inputs and leveraging our theoretical understanding of the underlying mathematics. Second, the results on log-uniform training distributions of operands and outcomes – faster learning and better performance – may extend to other arithmetic tasks, e.g. fine tuning LLM. Finally, mathematical tasks play a central role for Foundational Models for Science – large language models pre-trained on mathematics, and fine-tuned on specific fields, such as high energy physics, computational biology or astrophysics. Before they can do science, transformers must learn maths.

RELATED WORK

**Neural networks for arithmetic** were first proposed by Siu & Roychowdhury (1992), and recurrent models by Kalchbrenner et al. (2015), Zaremba et al. (2015) and Kaiser & Sutskever (2015). Recent research mostly focuses on fine-tuning LLM on arithmetic tasks, to solve math word problems (Meng & Rumshisky, 2019; Griffith & Kalita, 2021). See Lee et al. (2023) for a summary. As an alternative, Neural Arithmetic Logical Units (Trask et al., 2018; Mistry, 2023) learn exact computations that can generalize to any input, by constraining the weights of linear models to be close to 0, 1 or $-1$.

**The difficulty of learning arithmetic tasks** was discussed by many authors. Saxton et al. (2019), benchmarking mathematical tasks, observe that number theoretic operations, like factorization, are hard. Palamas (2017) further investigates the hardness of modular arithmetic. Dziri et al. (2023) note the difficulty of extending the promising results obtained by Lee et al. (2023) on the four operations to complex mathematical calculations or algorithms – GCD and Euclid's algorithm, here.

**The role of number representation** was discussed by Nogueira et al. (2021) and Charton (2021). **Grokking** was first described by Power et al. (2022). Liu et al. (2022) propose metrics to characterize it. Gromov (2023) provides an insightful analysis of grokking in feed-forward networks. Most prior work on **explainability in arithmetic transformers** tries to interpret model weights (Nanda et al., 2023; Zhong et al., 2023). Charton (2022a) conducts similar experiments for linear algebra.

## 2 EXPERIMENTAL SETTINGS

GCD calculations are framed as a supervised translation task. Problems (pairs of integers) are randomly sampled, represented as sequences of tokens, and used to train sequence-to-sequence transformers to translate input pairs into their GCD, by minimizing the cross-entropy between model predictions and the sequences representing correct solutions. Integers are encoded as sequences of digits in base $B$, preceded by a sign which also serves as a separator (Table 1). In base 10, the model translates $(8, 12)$, encoded as the sequence '+ 8 + 1 2', into its GCD, 4, encoded as '+ 4'. The choice of $B$ is a trade-off. Small bases result in longer sequences that are harder to learn, but use a small vocabulary that is easier to memorize. Composite bases allow for simple tests of divisibility: in base 10, divisibility by 2, 5 and 10 is decided by looking at the rightmost token in the sequence.

Transformers with 4 layers, 512 dimensions and 8 attention heads, using Adam (Kingma & Ba, 2014) are trained with a learning rate of $10^{-5}$ (no scheduling is needed) on batches of 256 examples. All inputs pairs are sampled uniformly between 1 and $M = 10^6$. All data is generated on the fly: different training epochs use different examples for the train and test set. After each epoch (300,000 examples), the models are evaluated on two test sets of 100,000 examples: a *natural test set* of uniformly sampled pairs $(a, b)$, and a *stratified test set* with GCD uniformly distributed between 1 and 100. In the natural set, small GCD are more common – we have $P(\gcd(a, b) = k) = \frac{6}{\pi^2 k^2}$ (Cesàro, 1883). The stratified set has about 1000 examples with GCD $k$ for $1 \leq k \leq 100$, and is generated by:

- sampling $k$, uniformly between 1 and 100,
- sampling $a$ and $b$, uniformly between 1 and $\frac{M}{k}$, such that $\gcd(a, b) = 1$, using rejection sampling,
- adding $(ka, kb)$ to the stratified test set.

These two test sets provide two measures of accuracy. **Model accuracy**, measured on the natural set, is the probability that the GCD of two random integers from 1 to $M$ is correctly predicted. Accuracy on the stratified test set is the **number of GCD correctly predicted** between 1 and 100. The size of the problem space ($10^{12}$ possible input pairs) guarantees minimal duplication between train and test set. All experiments are run on one NVIDIA V100 GPU with 32 GB of memory. The source code for these experiments can be found at `https://github.com/facebookresearch/GCD`.

Table 1: Encoding gcd(160,120) = 40, in base 2, 6, 10 and 30

| Base | Encoded input | Encoded output |
|------|---------------|----------------|
| 2 | [+,1,0,1,0,0,0,0,0,+,1,1,1,1,0,0,0] | [+,1,0,1,0,0,0] |
| 6 | [+,4,2,4,+,3,2,0] | [+,1,0,4] |
| 10 | [+,1,6,0,+,1,2,0] | [+,4,0] |
| 30 | [+,5,10,+,4,0] | [+,1,10] |

Table 2: Number of correct GCD under 100 and accuracy. Best of 6 experiments.

| Base | 2 | 3 | 4 | 5 | 6 | 7 | 10 | 11 | 12 | 15 |
|------|---|---|---|---|---|---|----|----|----|----|
| Correct GCD | 7 | 5 | 7 | 3 | 19 | 3 | 13 | 2 | 19 | 9 |
| Accuracy | 81.6 | 68.9 | 81.4 | 64.0 | 91.5 | 62.5 | 84.7 | 61.8 | 91.5 | 71.7 |
| Base | 30 | 31 | 60 | 100 | 210 | 211 | **420** | 997 | 1000 | 1024 |
| Correct GCD | 27 | 2 | 28 | 13 | 32 | 1 | **38** | 1 | 14 | 7 |
| Accuracy | 94.7 | 61.3 | 95.0 | 84.7 | 95.5 | 61.3 | **96.8** | 61.3 | 84.7 | 81.5 |

## 3 LEARNING THE GREATEST COMMON DIVISOR - BASE EXPERIMENTS

A model trained on pairs of positive integers under one million, encoded in base $B = 10$, correctly predicts $84.7\%$ of the examples in the natural test set, and 13 correct GCD under 100 (accuracy on the stratified test set). Performances vary with the encoding base: from $61.8\%$ accuracy and 2 correct GCD for base 11, to $96.8\%$ and 38 GCD for base 420 (Table 2). The best performances are achieved for composite bases (30, 60, 210 and 420), the worst for large primes. Learning is very fast: for base 30, the model achieves $90\%$ accuracy after 2 epochs (600,000 examples), and $93\%$ after 6. Model size has little impact on performance (Appendix B). For base 30, 1-layer transformers with 32 dimensions (less than 300,000 parameters) achieve $93.3\%$ accuracy. 24-layer models with 1024 dimensions (714 million parameters) achieve $93.4\%$. For base 31, accuracy is $61\%$ for all models.

These variations in model performance can be understood by looking at model predictions. Table 3 presents, for bases 2 and 10 and GCD up to 36, the most frequent model prediction for pairs with a given GCD (Pred), and its frequency in the stratified test set ($\%$) – detailed results for 6 bases and GCD up to 100 are in Appendix E.3). All frequencies are very close to $100\%$: for every test pair with GCD $k$, the model makes the same prediction $f(k)$. In other words, the model can tell whether two input pairs have the same GCD. Correct model predictions ($f(k) = k$) only happen for products of divisors of the base. In fact, all model predictions can be summarized in **three rules**:

(R1) **Predictions are deterministic.** The model predicts a unique value $f(k)$ for almost all ($99.9\%$) pairs of integers with GCD $k$. Predictions are correct when $f(k) = k$.
(R2) **Correct predictions are products of primes dividing B.** For base 2, they are 1, 2, 4, 8, 16, 32 and 64. For base 31, 1 and 31. For base 10, all products of elements from $\{1, 2, 4, 8, 16\}$ and $\{1, 5, 25\}$. For base 30, all products of $\{1, 2, 4, 8\}$, $\{1, 3, 9, 27\}$ and $\{1, 5, 25\}$.
(R3) **f(k) is the largest correct prediction that divides k.** For instance, $f(8) = 8$, and $f(7) = 1$, for base 2 and 10, but $f(15) = 5$ for base 10 and $f(15) = 1$ for base 2.

These results can be interpreted as follows. For prime bases, such as $B = 2$, an integer is divisible by $B^k$ iff its representation ends in $k$ zeroes. The model learns to "predict" GCD by counting the rightmost zeroes in its operands, $z_a$ and $z_b$, and predicting $B^z$ with $z = \min(z_a, z_b)$. This accounts for all observed results. For instance, it will correctly predict the GCD of $a = 8 = 1000_2$ and $b = 12 = 1100_2$ to be $2^2 = 4$, and incorrectly predict the GCD of $7 = 111_2$ and $14 = 1110_2$ to be

Table 3: Model predictions and their frequencies, for GCD 1 to 36. Correct predictions in bold face.

| | Base 2 | | Base 10 | | | Base 2 | | Base 10 | | | Base 2 | | Base 10 | |
|-----|--------|-----|---------|-----|-----|--------|-----|---------|-----|-----|--------|-----|---------|-----|
| GCD | Pred | % | Pred | % | GCD | Pred | % | Pred | % | GCD | Pred | % | Pred | % |
| 1 | **1** | 100 | **1** | 100 | 13 | 1 | 100 | 1 | 100 | 25 | 1 | 100 | **25** | 100 |
| 2 | **2** | 100 | **2** | 100 | 14 | 2 | 100 | 2 | 100 | 26 | 2 | 100 | 2 | 100 |
| 3 | 1 | 100 | 1 | 100 | 15 | 1 | 100 | 5 | 100 | 27 | 1 | 100 | 1 | 100 |
| 4 | **4** | 100 | **4** | 100 | 16 | **16** | 100 | **16** | 99.7 | 28 | 4 | 100 | 4 | 100 |
| 5 | 1 | 100 | **5** | 100 | 17 | 1 | 100 | 1 | 100 | 29 | 1 | 100 | 1 | 100 |
| 6 | 2 | 100 | 2 | 100 | 18 | 2 | 100 | 2 | 100 | 30 | 2 | 100 | 10 | 100 |
| 7 | 1 | 100 | 1 | 100 | 19 | 1 | 100 | 1 | 100 | 31 | 1 | 100 | 1 | 100 |
| 8 | **8** | 100 | **8** | 100 | 20 | 4 | 100 | **20** | 100 | 32 | **32** | 99.9 | 16 | 99.9 |
| 9 | 1 | 100 | 1 | 100 | 21 | 1 | 100 | 1 | 100 | 33 | 1 | 100 | 1 | 100 |
| 10 | 2 | 100 | **10** | 100 | 22 | 2 | 100 | 2 | 100 | 34 | 2 | 100 | 2 | 100 |
| 11 | 1 | 100 | 1 | 100 | 23 | 1 | 100 | 1 | 100 | 35 | 1 | 100 | 5 | 100 |
| 12 | 4 | 100 | 4 | 100 | 24 | 8 | 100 | 8 | 100 | 36 | 4 | 100 | 4 | 100 |

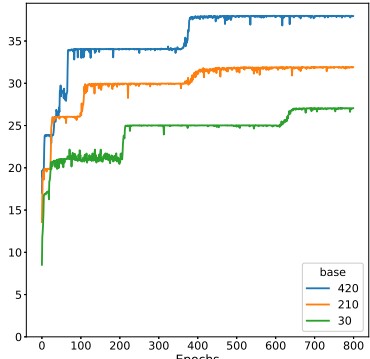

Figure 1: Correct GCD vs training time. Natural ($\frac{1}{k^2}$) distribution of GCD.

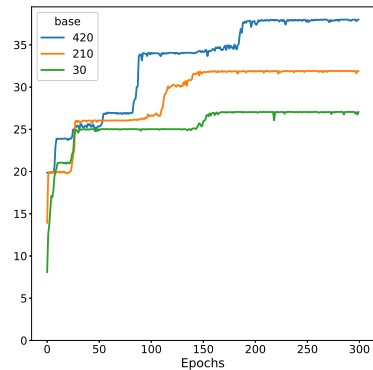

Figure 2: Correct GCD vs training time. 5% uniform, 95% natural GCD.

1. For composite bases, such as $B = 10$, an integer $a$ is divisible by $f$, such that $kf = B^n$, iff its $n$ rightmost digits are in $\{0, f, 2f \ldots (k-1)f\}$. The model learns to test the divisibility of its operands by comparing their $n$ rightmost digits with the $k$ possible values, and predict the largest $f$ that divides both operands. In practice, only divisibilities that can be tested by considering the two last digits in the representation are learned. For $B = 210$ divisibility by $4$ is learned, but divisibility by $8$ is not. For $B = 420$ divisibility by $16$ is learned, but not by $32$. The three rules also account for variations in model accuracy (computed on the natural test set) for different bases (see Appendix C).

**Learning GCD one prime power at a time.** Learning curves have a step-like shape (Figure 1), and GCD are learned in sudden batches. When the model learns a new power of a prime divisor of $B$, it also learns its products with already known GCD. For instance, for base 30, the model initially predicts $\{1, 2, 4\}$, $\{1, 3, 9\}$, $\{1, 5\}$ and their products: 17 GCD under 100. A first step happens around epoch 50, when the model learns 25 and the three associated multiples 50, 75 and 100 (21 GCD), a second around epoch 220, learning 8, 24, 40 and 72, and a third at epoch 660, learning 27 and 54, for a grand total of 27 correct GCD. The three rules hold at all times during training.

**Accelerating learning by balancing the distribution of GCD.** The distribution of GCD verifies $P(\gcd(a, b) = k) = \frac{6}{\pi^2 k^2}$ (Cesàro, 1883). As a result, large GCD are very rare in the training set, and learning them is very slow. This can be mitigated, and training accelerated, by adding a small proportion (5%) of uniformly sampled GCD to the training set: for $B = 30$, the model learns 25 GCD in 30 epochs, and 27 GCD in 175, vs 250 and 660 in the original experiments (Figure 2).

In these experiments, models only correctly calculate GCD that are products of divisors of the base, and the best accuracies are achieved for bases divisible by many small primes, e.g. 30, 210 or 420. Still, all models learn to cluster pairs of input integers according to their GCD, and output a unique prediction $f(k)$ for all pairs with GCD $k$. This is a non-trivial result and a significant achievement.

## 4 LARGE COMPOSITE BASES $B$ - GROKKING SMALL PRIMES

For large bases $B$, non-divisors of $B$ are sometimes learned after extended training. In one experiment with base 1000, the model predicts 13 GCD $\leq 100$ after 84 epochs: all products of $\{1, 2, 4, 8, 16\}$ and $\{1, 5, 25\}$. Then, the training loss is flat during 100 epochs, and it seems that the model is no longer learning anything. But then, the model starts predicting GCD 3, with an accuracy of $0.2\%$ at epoch 188, and $93\%$ at epoch 193 (despite only seeing 100,000 input pairs with GCD 3 during these 5 epochs). Multiples of 3 are then learned, and by epoch 220, the model predicts 22 GCD: all products of $\{1, 2, 4, 8, 16\}$, $\{1, 5, 25\}$ and $\{1, 3\}$. Model predictions still respect rules R1 and R3 (Appendix E.1 Table 20), and the **three rules** can be updated as follows:

(G1) **Prediction is deterministic.** All pairs with the same GCD are predicted the same, as $f(k)$.
(G2) **Correct predictions are products of primes divisors of B and small primes**.
(G3) **f(k) is the largest correct prediction that divides k.**

This phenomenon is related to grokking (Power et al., 2022). Table 4 presents results for 16 large bases, with models trained up to 1300 epochs. Grokking usually sets in late during training: for bases 625 and 4000, all products of divisors of $B$ are learned in 5 and 15 epochs, but it take 600 epochs for grokking (of 2 and 3) to happen. Primes and powers of primes are roughly grokked in order.

Table 4: **Predicted gcd, divisors and non-divisors of B.** Best model of 3. For non-divisors, the epoch learned is the first epoch where model achieves 90% accuracy for this GCD.

| Base | GCD predicted | Divisors predicted | Non-divisors (epoch learned) |
|---|---|---|---|
| $625 = 5^4$ | 6 | {1,5,25} | 2 (634) |
| 2017 | 4 | {1} | 2 (142), 3 (392) |
| $2021 = 43.47$ | 10 | {1,43}, {1,47} | 2 (125), 3 (228) |
| $2023 = 7.17^2$ | 16 | {1,7}, {1,17} | 3 (101), 2 (205), 4 (599) |
| $2025 = 3^4.5^2$ | 28 | {1,3, 9, 27, 81}, {1,5,25} | 2 (217), 4 (493), 8 (832) |
| $2187 = 3^7$ | 20 | {1,3,9,27,81} | 2 (86), 4 (315) , 5 (650) |
| $2197 = 13^3$ | 11 | {1,13} | 2 (62), 3 (170), 4 (799) |
| $2209 = 47^2$ | 8 | {1,47} | 2 (111), 3 (260), 9 (937) |
| $2401 = 7^4$ | 10 | {1,7,49} | 2 (39), 3 (346) |
| $2401 = 7^4$ | 14 | {1,7,49} | 3 (117), 2 (399), 4 (642) |
| $2744 = 2^3.7^3$ | 30 | {1,2,4,8,16,32}, {1,7,49} | 3 (543), 5 (1315) |
| $3125 = 5^5$ | 16 | {1,5,25} | 2 (46), 3 (130), 4 (556) |
| $3375 = 3^3.5^3$ | 23 | {1,3,9,27}, {1,5,25} | 2 (236), 4 (319) |
| $4000 = 2^5.5^3$ | 24 | {1,2, 4,8,16,32}, {1, 5, 25 } | 3 (599) |
| $4913 = 17^3$ | 17 | {1,17} | 2 (54), 3 (138), 4 (648), 5 (873) |
| $5000 = 2^3.5^4$ | 28 | {1,2,4,8,16,32}, {1,5,25} | 3 (205), 9 (886) |
| $10000 = 2^4.5^4$ | 22 | {1,2,4,8,16}, {1,5,25} | 3 (211) |

Learning curves (Appendix E.1 Figure 5) retain their usual step-like shape: long periods of stagnation followed by sudden drops in the loss, and rises in accuracy, as new GCD are learned. Because it helps learn small GCD, grokking boosts model accuracy (from 63% to 91% for $B = 2023$), but overall the number of correct GCD remains low (under 30 for all large bases).

**Balancing outcomes.** The technique proposed in section 3 to accelerate learning (adding a small amount of uniformly distributed GCD to the training set) does not apply to larger bases (Appendix E.3 Table 21). However, the unbalanced distribution of GCD can be corrected by sampling from a log-uniform distribution – so that $P(\gcd(a,b) = k) = \frac{C}{k}$ instead of $\frac{C}{k^2}$ – as follows:

- Sample $k$ between 1 and 100, with probability $P(k) = \frac{C}{k}$, with $\frac{1}{C} = \sum_{i=1}^{100} \frac{1}{i}$.
- Sample $a$ and $b$ uniformly from 1 to $\frac{M}{k}$, such that $\gcd(a,b) = 1$.
- Add $(ak, bk)$ to the training set.

A log-uniform training distribution of GCD helps the model learn new non-divisors of $B$ for 9 bases out of 35 (Table 5). For $B = 211$, primes up to 7 are learned. For $B = 10000$, 7, 9, 13 and 27 are learned, bringing the number of correct GCD to 62, our best result so far. For $B = 30$, a counter-intuitive situation prevails: instead of small primes, the model learns $B - 1$ and $B + 1$.

Table 5: **Log-uniform vs natural outcomes.** Best model of 3, trained for 700 epochs. Non-divisors in bold.

| Base | Natural # GCD | Log-uniform outcomes # GCD | New divisors learned | Base | Natural # GCD | Log-uniform outcomes # GCD | New divisors learned |
|---|---|---|---|---|---|---|---|
| 2 | 7 | 7 | - | 997 | 1 | 1 | - |
| 3 | 5 | 5 | - | 1000 | 22 | 31 | **9**, 32, 64 |
| 4 | 7 | 7 | - | 2017 | 4 | 6 | **9** |
| 5 | 3 | 3 | - | 2021 | 10 | 10 | - |
| 6 | 19 | 20 | 64 | 2023 | 16 | 11 | - |
| 7 | 3 | 3 | - | 2025 | 28 | 28 | - |
| 10 | 13 | 14 | 32 | 2187 | 20 | 20 | - |
| 11 | 2 | 2 | - | 2197 | 11 | 11 | - |
| 12 | 19 | 20 | 81 | 2209 | 8 | 8 | - |
| 15 | 9 | 10 | 81 | 2401 | 14 | 16 | **5** |
| 30 | 25 | 36 | 16, **29, 31** | 2744 | 29 | 21 | - |
| 31 | 2 | 2 | - | 3125 | 16 | 16 | - |
| 60 | 28 | 33 | 27, 32, 64 | 3375 | 23 | 21 | - |
| 100 | 13 | 15 | 32, 64 | 4000 | 25 | 31 | **9**, 64 |
| 210 | 32 | 32 | - | 4913 | 17 | 9 | - |
| 211 | 1 | 18 | **2,3,4,5,7** | 5000 | 28 | 30 | 64 |
| 420 | 38 | 47 | **13, 49** | 10000 | 22 | 40 | **7, 9**, 32 |
| 625 | 6 | 9 | **4** | 10000 | 22 | 62 | **7, 9, 13, 27**, 32, 64 |

## 5 LEARNING FROM LOG-UNIFORM OPERANDS

In all experiments so far, all pairs in the training sets are uniformly sampled between 1 and $10^6$. As a result, models are mostly trained from examples with large operands. $90\%$ of operands are larger than 100,000, and small instances, like $\gcd(6, 9)$, are almost never encountered. This contrast with the way we are taught, and teach, arithmetic. We usually insist that small examples should be mastered, and sometimes memorized, before larger instances, like $\gcd(102370, 102372)$ can be tackled.

In this section, I sample training pairs from a log-uniform distribution, by uniformly sampling real numbers $0 \le x \le \log M$, computing $e^x$ and rounding to the nearest integer. In this setting, the training set has as many 1-digit as 6-digit operands. In $3\%$ of training example, both operands are smaller than 10, and in $11\%$ of examples, both are smaller than 100. This presents the model with many simple examples that it can memorize, just like children rote learn multiplication and addition tables. This is different from curriculum learning: the distribution of operands does not change during training. Also, the log-uniform sampling only applies to the training set (the test sets are unaffected), and it has no impact on the distribution of outcomes.

Table 6: **Accuracy and correct GCD (up to 100), log-uniform operands.** Best of three models, trained for 1000 epochs (300M examples). All models are tested on 100,000 pairs, uniformly distributed between 1 and $10^6$.

| Base | Accuracy | Correct GCD | Base | Accuracy | GCD | Base | Accuracy | GCD |
|------|----------|-------------|------|----------|-----|------|----------|-----|
| 2 | 94.4 | 25 | 60 | 98.4 | 60 | 2025 | 99.0 | 70 |
| 3 | 96.5 | 36 | 100 | 98.4 | 60 | 2187 | 98.7 | 66 |
| 4 | 98.4 | 58 | 210 | 98.5 | 60 | 2197 | 98.8 | 68 |
| 5 | 97.0 | 42 | 211 | 96.9 | 41 | 2209 | 98.6 | 65 |
| 6 | 96.9 | 39 | 420 | 98.1 | 59 | **2401** | **99.1** | **73** |
| 7 | 96.8 | 40 | 625 | 98.2 | 57 | 2744 | 98.9 | 72 |
| 10 | 97.6 | 48 | 997 | 98.3 | 64 | 3125 | 98.6 | 65 |
| 11 | 97.4 | 43 | 1000 | 99.1 | 71 | 3375 | 98.8 | 67 |
| 12 | 98.2 | 55 | 1024 | 99.0 | 71 | 4000 | 98.7 | 66 |
| 15 | 97.8 | 52 | 2017 | 98.6 | 63 | 4913 | 98.2 | 57 |
| 30 | 98.2 | 56 | 2021 | 98.6 | 66 | 5000 | 98.6 | 64 |
| 31 | 97.2 | 44 | 2023 | 98.7 | 65 | 10000 | 98.0 | 56 |

Training from log-uniform operands greatly improves performance (Table 6). Accuracy for all bases is between 94 and $99\%$, compared to 61 and $97\%$ with uniform operands. **For base 2401, the number of correct GCD is 73, our best result so far**. For base 10, the number of correct GCD is 48 (vs 13). Learning is accelerated: for base 10, GCD $1, 2, 4$ and 5 are learned as early as epoch 3, 3 and 8 by epoch 25, 7 and 9 by epoch 220 and 11 by epoch 750.

As before, large bases perform better. All models with $B \le 420$ have an accuracy over $98\%$ and correctly predict more than 55 GCD under 100. The divisors or $B$ are learned first, then, small powers of primes are grokked, roughly in order. After training, models have learned to predict all primes up to a certain value, some of their small powers, and all associated products. All primes up to 5 are learned for base 2, up to 11 for base 10, up to 17 for base 100, and up to 23 for base 1024. For base 1024, 2401, and 2744, only 27 GCD are incorrectly predicted:

- the 16 primes from 29 and 97, all predicted as 1,
- small multiples of these primes: products of 2 and $29, 31, 37, 41, 43$ and 47, predicted as 2, and products of 3 and 29 and 31, predicted as 3,
- powers of small primes: $49 = 7^2$, predicted as 7, and $81 = 3^4$, predicted as 27.
- small multiples of these: $98 = 49 * 2$, predicted as 14.

The three rules with grokking (G1 to G3) still apply: predictions are deterministic, for a pair $(a, b)$ with GCD $k$, the model predicts the largest correctly predicted GCD that divides $k$.

Learning curves retain their step-like shape, but they are more noisy, and smoother (see Appendix E.2): transitions now span several epochs, and each new prime takes more examples to be fully learned. While the model During training, while the model learns a new divisor, rules G1 and G3 are temporarily violated. During a few epochs, model predictions are split between the old and the new value (e.g. between 7 and 49 when the model is learning 49). This situation, rarely observed in previous experiments, is common with log-uniform operands.

Table 7: **Accuracy and correct GCD, log-uniform operands and outcomes.** Best model of 3.

| Base | Accuracy | Correct GCD | Base | Accuracy | GCD | Base | Accuracy | GCD |
|------|----------|-------------|------|----------|-----|------|----------|-----|
| 2 | 16.5 | 17 | 60 | 96.4 | 75 | **2025** | **97.9** | **91** |
| 3 | 93.7 | 51 | 100 | 97.1 | 78 | 2187 | 97.8 | 91 |
| 4 | 91.3 | 47 | 210 | 96.2 | 80 | 2197 | 97.6 | 90 |
| 5 | 92.2 | 58 | 211 | 95.3 | 67 | 2209 | 97.6 | 87 |
| 6 | 95.2 | 56 | 420 | 96.4 | 88 | 2401 | 97.8 | 89 |
| 7 | 93.0 | 63 | 625 | 96.0 | 80 | 2744 | 97.6 | 91 |
| 10 | 94.3 | 65 | 997 | 97.6 | 83 | 3125 | 97.7 | 91 |
| 11 | 94.5 | 57 | **1000** | **97.9** | **91** | 3375 | 97.6 | 91 |
| 12 | 95.0 | 70 | 1024 | 98.1 | 90 | 4000 | 97.3 | 90 |
| 15 | 95.4 | 62 | 2017 | 97.6 | 88 | 4913 | 97.1 | 88 |
| 30 | 95.8 | 72 | 2021 | 98.1 | 89 | 5000 | 97.1 | 89 |
| 31 | 94.4 | 64 | 2023 | 97.5 | 88 | 10000 | 95.2 | 88 |

**Log-uniform outcomes.** Balancing the distribution of GCD by making it log-uniform, as described in section 4, together with log-uniform operands, brings another large improvement in performance (Table 7). After 1000 epochs, **all models with B larger than 1000 predict 87 to 91 GCD**: all primes up to 53 and all composite numbers up to 100. These are our best results. They can be marginally improved by training models from an inverse square root distribution of outcomes (Appendix D.1). Note the low accuracy for base 2: with log-uniform outcomes, the model fails to learn GCD 1, for lack of examples.

## 6   LEARNING FROM UNIFORM OUTCOMES

Log-uniform distributions of outcomes improve model performance by reducing the imbalance between small and large GCD in the training set. It is therefore tempting to push this logic further, and train models from a uniform distribution of GCD and operands, i.e. sample the training set like the stratified test set from Section 2. Figure 3 presents learning curves for three models using base 10. Model accuracy (measured on the natural test set) seems to vary randomly, and the test loss is flat. Yet, the number of correct GCD is stable over time, and increases in steps, from 10 to 17, in line with the results from section 3 (13 GCD are learned). Something is learned despite the flat loss.

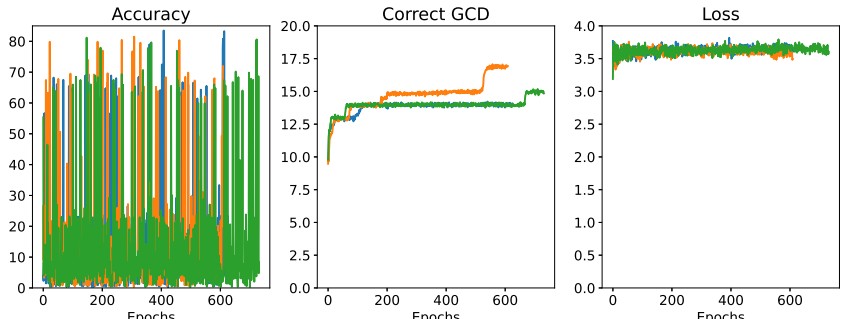

Figure 3: **Learning curves for B=10. Uniform outcomes and operands.** 3 different seeds.

Table 8 presents the most common model predictions, and their frequencies, for all GCD up to 20. At first glance, predictions seem chaotic. At epoch 266, the model achieves 81% accuracy, and correctly predicts 14 GCD: 1, 2, 5, 8, 20, 32, 40, 44, 48, 50, 64, 75, 80 and 100. One epoch later, accuracy is down to 6%, the model still predicts 14 GCD: 4, 8, 10, 16, 40, 50, 55, 60, 64, 66, 75, 80, 95 and 100, half of the correct GCD have changed! After another epoch, accuracy is 4% and the model predicts 4, 20, 25, 26, 30, 32, 40, 48, 50, 55, 64, 73, 80, 88 and 100. Again, half the correct GCD have changed.

As in previous experiments, frequencies are close to 100%: the model makes a unique prediction $f(k)$ for all pairs with GCD $k$, with the notable exception of epoch 267 where model predictions for 1, 3 ... are split (almost evenly) between 11 and 19. Model predictions cluster by classes of GCD: all elements in class $C_1 = \{1, 3, 7, 9, 11, 13, 17, 19\}$ are predicted as 1 at epoch 266, 19 at epoch 267, 73 at epoch 268, and so on. The same pattern appears for classes $C_2 = \{2, 6, 14, 18\}$, $C_4 = \{4, 12\}$

Table 8: **Prediction for base 10 - uniform operands and outcomes.** Most common prediction for GCD 1 to 20, and frequency, for successive epochs. Correct predictions are in bold

| | Epoch 266 | | Epoch 267 | | Epoch 268 | | Epoch 269 | | Epoch 270 | | Epoch 580 | | Epoch 581 | |
| | Pred | % | Pred | % | Pred | % | Pred | % | Pred | % | Pred | % | Pred | % |
|---|---|---|---|---|---|---|---|---|---|---|---|---|---|---|
| 1 | **1** | 100 | 19 | 54 | 73 | 100 | 7 | 100 | 13 | 100 | **1** | 98 | 77 | 99 |
| 2 | **2** | 100 | 66 | 100 | 26 | 100 | 62 | 100 | 66 | 100 | 22 | 93 | 22 | 99 |
| 3 | 1 | 100 | 19 | 52 | 73 | 100 | 7 | 100 | 13 | 100 | 1 | 99 | 77 | 99 |
| 4 | 44 | 91 | **4** | 100 | **4** | 100 | 44 | 100 | **4** | 100 | **4** | 100 | **4** | 100 |
| 5 | **5** | 100 | 55 | 100 | 55 | 100 | 55 | 100 | **5** | 100 | **5** | 100 | **5** | 100 |
| 6 | 2 | 100 | 66 | 100 | 26 | 200 | 62 | 100 | 66 | 100 | 22 | 93 | 22 | 99 |
| 7 | 1 | 100 | 19 | 62 | 73 | 100 | **7** | 100 | 13 | 100 | 1 | 99 | 77 | 99 |
| 8 | **8** | 99 | **8** | 100 | 88 | 100 | **8** | 100 | **8** | 100 | 88 | 100 | 88 | 99 |
| 9 | 1 | 100 | 19 | 53 | 73 | 100 | 7 | 100 | 13 | 100 | 1 | 99 | 77 | 99 |
| 10 | 70 | 70 | **10** | 100 | 30 | 99 | 70 | 100 | 70 | 100 | 30 | 100 | 70 | 100 |
| 11 | 1 | 100 | 19 | 57 | 73 | 100 | 7 | 100 | 13 | 100 | 1 | 98 | 77 | 99 |
| 12 | 44 | 91 | 4 | 100 | 4 | 100 | 44 | 100 | 4 | 100 | 4 | 100 | 18 | 22 |
| 13 | 1 | 100 | 19 | 55 | 73 | 100 | 7 | 100 | **13** | 100 | 1 | 98 | 77 | 99 |
| 14 | 2 | 100 | 66 | 100 | 26 | 100 | 62 | 100 | 66 | 100 | 22 | 92 | 22 | 99 |
| 15 | 5 | 100 | 55 | 100 | 55 | 100 | 55 | 100 | 5 | 100 | 5 | 100 | 5 | 100 |
| 16 | 48 | 97 | **16** | 84 | 48 | 99 | 48 | 99 | **16** | 98 | 48 | 98 | 48 | 78 |
| 17 | 1 | 100 | 19 | 54 | 73 | 100 | 7 | 100 | 13 | 100 | 1 | 99 | 77 | 100 |
| 18 | 2 | 100 | 66 | 100 | 26 | 100 | 62 | 100 | 66 | 100 | 22 | 93 | 22 | 99 |
| 19 | 1 | 100 | **19** | 53 | 73 | 100 | 7 | 100 | 13 | 100 | 1 | 99 | 77 | 99 |
| 20 | **20** | 100 | 60 | 100 | **20** | 98 | **20** | 100 | **20** | 53 | **20** | 100 | **20** | 100 |

and $C_5 = \{5, 15\}$, i.e. pairs of integers both divisible by 2, 4, and 5, that would have been predicted as 2, 4, and 5 by the base 10 model from section 3. In other words, the model learns to cluster input pairs into classes having a common divisor (a product of divisors of 10), just like it did in section 3, but instead of predicting the smallest (and most common) element in each class, it predict a different element at every epoch. This can be summarized into **three rules with uniform outcomes**:

(U1) **Predictions are mostly deterministic.** At a given epoch, the model usually predicts a unique value $f(k)$ for a given GCD $k$. In rare cases, the model makes 2 or 3 predictions.

(U2) **Classes of multiples of products of prime divisors of B are predicted the same.** For base 10, some classes are $C_1 = \{1, 3, 7, 9, 11, 13, 17, 19 \ldots\}$, $C_2 = \{2, 6, 14, 18, 22, 26, 34, 38 \ldots\}$, $C_4 = \{4, 12, 24, 36, 44, 52, \ldots\}$ and $C_5 = \{5, 15, 35, 55 \ldots\}$.

(U3) **For each class, the model prediction is an element of the class.** Prediction varies from one epoch to the next, but the number of correct GCD is stable over time: it is the number of classes, which increases as the model learns new divisors of $B$.

The three rules explain the variations in the accuracy curve: since $61\%$ of examples in the natural test set have GCD 1, accuracy jumps by $61\%$ every time class $C_1$ is predicted as 1. Rule U3, on the other hand, accounts for the step-shaped learning curve for correct GCD.

These results shed light on the learning process and the role of the distribution of outcomes. During training, all models, regardless of outcome distribution, learn to partition their input pairs into classes, with GCD multiples of a product of divisors of the base (or small primes when grokking happens), i.e. for base 10, multiples of 2, 4, 5, 10, 20, and a default class associated to 1. The model makes a unique prediction for all pairs in a class. When the distribution of outcomes is unbalanced, this prediction is the smallest element in the class, which happens to be the most common. When outcomes are uniformly distributed, a different element of the class is predicted at every epoch, somewhat randomly: the model becomes less explainable.

**Base 1000, grokking and loss of determinism**. Models with base 1000, trained on uniform operands and outcomes, undergo a similar learning process (see Appendix D.3) during the first 400 training epochs. Grokking sets in around epoch 200. Multiples of 11, 22, 44, 55 and 88 are learned around epoch 220, then multiples of 3 by epoch 260 and of 7 by epoch 400. At this point, 41 GCD are correctly predicted. Note that grokking no longer happens in order: 11 is learned before 3.

During the grokking phase, a new phenomenon develops. As new primes are grokked and more classes are created, model predictions for each class become less deterministic. Instead of predicting a unique value for each class at each epoch, the model now "hesitates" between several values, and the frequency of the most common prediction goes down. By epoch 400, for the class $C_1$, the model makes 18 different predictions with frequencies ranging from $2\%$ to $13\%$ (Table 15 in Appendix D.3). Model predictions are no longer explainable, and the three rules are not respected.

Interestingly, GCD continue to be learned under this new regime, starting with the largest (i.e. the smallest classes of multiples). By epoch 740, 95 GCD under 100 are correctly predicted. The worst performance is achieved for small GCD: 43, 74 and $85\%$ correct predictions for GCD 1, 2 and 3. Appendix D.4 presents results for larger bases, where up to 99 GCD under 100 are learned.

## 7    DISCUSSION

**Can transformers learn the greatest common divisor?** With enough examples and appropriate adjustment of their training distribution, they can. Models leveraging large composite bases, and trained on log-uniform operands and outcomes predict over 90 of the 100 first GCD. Models trained on uniform outcomes predict 95 GCD. However, the experiments from section 3 show the limits of naive, benchmark-based evaluations on arithmetic tasks: high accuracies ($95\%$) can be achieved, on held-out test sets of of random examples, by models that only predict a handful of GCD.

**The approach to explainability** presented in this paper differs from most works on the subject. Instead of looking at model parameters, I engineer experiments that reveal the algorithms that the model is implementing. It is often repeated that transformers are incomprehensible black-boxes, that sometimes confabulate and often fail in unpredictable ways. Here, model predictions can be fully characterized by a small number of rules. This is a promising direction for future research.

Experiments indicate that **transformers learn a sieve algorithm for computing GCD**. The model first learns divisibility by products of divisors of the base, which can be tested by looking at the last digits of a number, or counting its rightmost zeroes. Using these rules, the model clusters its input pairs into classes of multiples of divisors of the base, and predicts the GCD as the minimum for the class. All GCD corresponding to products of divisors of $B^2$ are learned this way. At the end of this phase, in base 2, the model correctly predicts $1, 2, 4, 8, 16$ and $32$.

As training proceeds, new prime divisors are learned (grokked) in order. They are all prime because multiples of previous divisors were learned already, i.e. the model functions like a sieve. Every time a new divisor $p$ is learned, all existing classes are split between multiples and non-multiples of $p$. In base 2, once the model learns divisibility by 3, six new classes are created: multiples of 3, 6, 12, 24, 48 and 96 (splitted from $1, 2, 4, 8, 16$ and $32$. This accounts for the steps observed in the learning curves. A GCD is correctly predicted once all the powers of primes dividing it are learned. Eventually, all GCD will be learned this way.

Experiments with uniform outcomes suggest that **an unbalanced training distribution of GCD is needed** for this algorithm to succeed, because it causes each class to be predicted by its smallest, and most common, member (the correct GCD), and it guarantees that primes are learned in order. Interestingly, this algorithm is not related to Euclid's algorithm. Note also that it is not specific to transformers: Appendix D.5 shows that similar results can be achieved with LSTM and GRU.

Another important finding is **the role of training distributions**. All models are tested on sets with uniform operands, but the best results are achieved with a log-uniform distribution of operands and outcomes in the training set. This may come as a surprise, since many authors observed that evaluating a model out of its training distribution has a negative impact on performance. The existence of special training distributions, that allow for faster learning and more robust models (with respect to out-of-distribution generalization) was already observed for linear algebra (Charton, 2022a).

A log-uniform distribution of operands strikes a balance between memorization and generalization, and helps models learn hard instances by memorizing easier cases. This is related to curriculum learning, but avoids catastrophic forgetting, because the training distribution never changes. These observations may apply to other arithmetic tasks. On the other hand, a log-uniform distribution of outcomes helps learning by enforcing a better representation of large GCD in the training set, a classical recipe in machine learning (calssifiers are often trained on balanced datasets). The counter-intuitive result is that a perfectly balanced, uniform training distribution set degrades performance by preventing the model from learning small GCD, and breaking model explainability.

**Is it really grokking?** Power et al. (2022) define grokking as "generalization far after overfitting." In all experiments, training and test data are generated on the fly from a very large problem space. No overfitting can happen, and the classical pattern of grokking, train accuracy dropping, and validation accuracy catching up after a long time, will not occur. The similarity with grokking lies in the sudden change in accuracy after a long stagnation of the training loss.

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

APPENDIX

## A  RATIONAL ARITHMETIC WITH TRANSFORMERS

In these experiments, transformers are trained to perform five arithmetic operations on positive rational numbers:

- comparison: given four positive integers $a, b, c$ and $d$, predict whether $\frac{a}{b} < \frac{c}{d}$.
- Integer division: given two integers $a$ and $b$, predict the integer $\lfloor \frac{a}{b} \rfloor$.
- Addition: given four integers $a, b, c$ and $d$, predict the sum $\frac{a}{b} + \frac{c}{d}$, in lowest terms.
- Multiplication: given four integers $a, b, c$ and $d$, predict the product $\frac{a}{b} \times \frac{c}{d}$, in lowest terms.
- Simplification: given two integers $a$ and $b$, predict the lowest term representation of $\frac{a}{b}$, i.e. $\frac{c}{d}$ with $c = \frac{a}{\gcd(a,b)}$ and $d = \frac{b}{\gcd(a,b)}$.

For the comparison, addition and multiplication tasks, all integers $a, b, c$ and $d$ are uniformly sampled between 1 and $M$ ($M$=100,000 or 1,000,000).

For the simplification task, 3 integers $m, n, p$ are uniformly sampled between 1 and $M$, I let $a = \frac{pm}{\gcd(m,n)}$ and $b = \frac{pn}{\gcd(m,n)}$ and the model is tasked to predict $a$ and $b$.

For the integer division task, 3 integers $m, n, p$ are uniformly sampled between 1 and $M$, with $m < n$, I let $a = pn + m$ and $b = n$, and the model is tasked to predict $p = \lfloor \frac{a}{b} \rfloor$.

All integers are encoded as sequences of digits in base $B$ (see section 2). Sequence to sequence transformers with 4 layers, 512 dimensions and 8 attention heads are trained to minimize a cross-entropy loss, using Adam with learning rate $10^{-4}$, inverse square root scheduling, linear warmup over $10,000$ optimization steps, and a batch size of 256. After each epoch (300,000 examples), models are tested on 100,000 random examples.

Comparison is learned to very high accuracy, and integer division to some extent. On the other hand, the three tasks involving GCD calculations (simplification, addition and multiplication) are not learned (Table 9).

Table 9: **Rational arithmetic with transformers. Accuracy of trained models** Best of 3 models, trained for 1000 to 1500 epochs.

| Base | Comparison M=$10^5$ | M=$10^6$ | Integer division M=$10^5$ | M=$10^6$ | Simplification M=$10^5$ | M=$10^6$ | Addition M=$10^5$ | M=$10^6$ | Multiplication M=$10^5$ | M=$10^6$ |
|---|---|---|---|---|---|---|---|---|---|---|
| 10 | 100 | 100 | 21.2 | 2.4 | 0.14 | 0.02 | 0 | 0 | 0 | 0 |
| 30 | 99.9 | 100 | 14.2 | 2.2 | 0.21 | 0.02 | 0 | 0 | 0 | 0 |
| 31 | 99.9 | 100 | 14.3 | 2.4 | 0.02 | 0 | 0 | 0 | 0 | 0 |
| 1000 | 100 | 99.9 | 8.8 | 0.7 | 0.09 | 0.01 | 0 | 0 | 0 | 0 |

## B  MODEL SCALING FOR THE BASE EXPERIMENTS

Section 3 presents results for 4-layer transformers with 512 dimensions and 8 attention heads. In this section, I experiment with very small models (down to 1 layer and 32 dimensions), and very large ones (up to 24 layers and 1024 dimensions). Note: in Tables 10 and 11, the number of trainable parameters are indicated for base 10, they will be larger for larger bases, because larger vocabularies increase the number of parameters in the embedding and decoding layers.

Table 10 presents accuracies for models with one layer, 8 attention heads, and 32 to 512 dimensions. These models have 3 to 100 times less parameters that the 4-layer baseline, but there is no significant change in trained model accuracy for 12 different bases.

Table 11 presents results for models from 6 to 24 layers, symmetric (same number of layers in the encoder and decoder), or asymmetric (using a one-layer encoder or decoder). The dimensions are 512, 640, 768 and 1024 for 6, 8, 12, and 24 layers, and the dimension-to-attention-heads ratio is kept constant at 64 (i.e.there are 8, 10, 12 and 24 attention heads respectively). Again, model size has no significant impact on accuracy.

Overall, these scaling experiments suggest that trained model performance is stable over a wide range of model size (300 thousands to 700 millions parameters). These results are strikingly different from what is commonly observed in Natural Language Processing, where very small transformers (under a few million parameters) cannot learn, and accuracy improves with model size.

Table 10: **Model accuracies for different dimensions and numbers of parameters.** All models have one layer and 8 attention heads. Parameter counts for base 10.

| Base | 512 dimensions 11.6M | 256 dim. 4.0M | 128 dim. 1.7M | 64 dim. 0.6M | 32 dim. 0.3M | 4-layer baseline 33.7M |
|---|---|---|---|---|---|---|
| 2 | 81.3 | 81.4 | 81.4 | 81.4 | 81.2 | 81.6 |
| 3 | 68.8 | 68.9 | 68.7 | 68.8 | 68.7 | 68.9 |
| 4 | 81.4 | 81.4 | 81.4 | 81.4 | 81.4 | 81.4 |
| 5 | 64.0 | 63.7 | 63.8 | 63.7 | 63.8 | 64.0 |
| 6 | 91.3 | 91.3 | 91.1 | 91.1 | 90.7 | 91.5 |
| 7 | 62.5 | 62.4 | 62.5 | 62.5 | 62.5 | 62.5 |
| 10 | 84.4 | 84.3 | 84.3 | 84.4 | 84.2 | 84.7 |
| 11 | 61.7 | 61.7 | 61.7 | 61.9 | 61.7 | 61.8 |
| 12 | 91.4 | 91.4 | 91.3 | 91.3 | 91.1 | 91.5 |
| 15 | 71.6 | 71.6 | 71.5 | 71.5 | 71.4 | 71.7 |
| 30 | 94.6 | 93.8 | 93.5 | 93.7 | 93.3 | 94.7 |
| 31 | 61.3 | 61.3 | 61.2 | 61.3 | 61.3 | 61.3 |

Table 11: **Model accuracies for different depths and number of parameters (in millions).** 1 and 6 layer models have 512 dimensions and 8 heads, 8-layer have 640 dimensions and 10 heads, 12-layer 768 dimensions and 12 heads, 24-layer models have 1024 dimensions and 16 heads. The largest base 2 and 3 models could not run on one 32GB GPU. All model parameters for base 10.

| Base | 1/6 32.5 | 6/1 27.3 | 6/6 48.3 | 1/8 59.1 | 8/1 48.4 | 8/8 97.1 | 1/12 117.1 | 12/1 94.7 | 12/12 204.8 | 1/24 387.4 | 24/1 313.3 | 24/24 713.8 |
|---|---|---|---|---|---|---|---|---|---|---|---|---|
| 2 | 81.3 | 81.3 | 81.4 | 81.5 | 81.4 | 81.3 | 81.3 | 81.3 | 81.4 | - | 81.4 | - |
| 3 | 68.7 | 68.8 | 68.7 | 68.8 | 68.9 | 69.0 | 68.9 | 68.8 | 68.8 | 68.8 | 68.6 | - |
| 4 | 81.3 | 81.4 | 81.4 | 81.4 | 81.4 | 81.6 | 81.4 | 81.4 | 81.4 | 81.5 | 81.4 | 81.3 |
| 5 | 63.8 | 63.8 | 63.7 | 63.8 | 63.6 | 63.7 | 63.7 | 63.7 | 63.6 | 63.9 | 63.7 | 63.6 |
| 6 | 91.3 | 91.1 | 91.3 | 91.3 | 91.4 | 91.3 | 91.3 | 91.0 | 91.0 | 91.3 | 91.0 | 90.9 |
| 7 | 62.6 | 62.6 | 62.4 | 62.5 | 62.4 | 62.6 | 62.5 | 62.4 | 62.4 | 62.4 | 62.3 | 62.2 |
| 10 | 84.3 | 84.2 | 84.4 | 84.7 | 84.4 | 84.5 | 84.4 | 84.4 | 83.4 | 84.5 | 83.4 | 83.3 |
| 11 | 61.8 | 61.7 | 61.6 | 61.7 | 61.8 | 61.7 | 62.0 | 61.6 | 61.7 | 61.7 | 61.6 | 61.6 |
| 12 | 91.4 | 91.3 | 91.3 | 91.4 | 91.5 | 91.4 | 81.4 | 91.2 | 91.2 | 91.4 | 91.3 | 91.2 |
| 15 | 71.5 | 71.5 | 71.4 | 71.5 | 71.5 | 71.5 | 71.4 | 71.5 | 71.5 | 71.5 | 70.6 | 71.4 |
| 30 | 94.6 | 93.4 | 93.5 | 94.7 | 93.6 | 93.6 | 94.7 | 93.6 | 93.6 | 93.5 | 93.4 | 93.4 |
| 31 | 61.2 | 61.2 | 61.3 | 61.2 | 61.3 | 61.2 | 61.4 | 61.2 | 61.3 | 61.4 | 61.3 | 61.1 |

## C  THEORETICAL VALUES OF ACCURACY

In this section, I compute a theoretical accuracy for models from section 3 that follow the three rules, assuming that all products of prime divisors of $B$ are correctly predicted. The distribution of the GCD of random uniform positive integers verifies: $P(\gcd(a,b) = k) = \frac{6}{\pi^2 k^2}$ (Cesàro, 1883).

Therefore, if $B = p^k$, with $p$ prime, theoretical model accuracy is

$$\mathcal{A}(p^k) = \mathcal{A}(p) = \frac{6}{\pi^2} \sum_{i=0}^{\infty} \frac{1}{p^{2i}} = \frac{6}{\pi^2} \frac{p^2}{p^2 - 1},$$

if $B = p^k q^l$, $\mathcal{A}(B) = 1 - \frac{\pi^2}{6}(1 - \mathcal{A}(p))(1 - \mathcal{A}(q))$,

if $B = p^k q^l r^m$, $\mathcal{A}(B) = 1 - \frac{\pi^4}{36}(1 - \mathcal{A}(p))(1 - \mathcal{A}(q))(1 - \mathcal{A}(r))$, and so on.

Table 12: Theoretical accuracy, accuracy and number of correct GCD under 100. Best of 6 experiments.

| Base | 2 | 3 | 4 | 5 | 6 | 7 | 10 | 11 | 12 | 15 |
|---|---|---|---|---|---|---|---|---|---|---|
| Theoretical accuracy | 81.1 | 68.4 | 81.1 | 63.3 | 90.2 | 62.1 | 88.6 | 61.3 | 90.2 | 80.3 |
| Accuracy | 81.6 | 68.9 | 81.4 | 64.0 | 91.5 | 62.5 | 84.7 | 61.8 | 91.5 | 71.7 |
| Correct GCD | 7 | 5 | 7 | 3 | 19 | 3 | 13 | 2 | 19 | 9 |

| Base | 30 | 31 | 60 | 100 | 210 | 211 | **420** | 997 | 1000 | 1024 |
|---|---|---|---|---|---|---|---|---|---|---|
| Theoretical accuracy | 94.1 | 60.9 | 94.1 | 88.6 | 96.3 | 60.8 | 96.3 | 60.8 | 88.6 | 81.1 |
| Accuracy | 94.7 | 61.3 | 95.0 | 84.7 | 95.5 | 61.3 | **96.8** | 61.3 | 84.7 | 81.5 |
| Correct GCD | 27 | 2 | 28 | 13 | 32 | 1 | **38** | 1 | 14 | 7 |

Table 12 compares theoretical accuracies with empirical observations. Best model performances may be higher than theory, because of sampling errors in the test set, or lower than theory when some powers of prime divisors of $B$ have not been learned.

# D  ADDITIONAL EXPERIMENTS

## D.1  EXPERIMENTS WITH OUTCOME DISTRIBUTIONS

The results at the end of section 5 demonstrate that training from a log-uniform distribution of GCD ($P(\text{gcd} = k) = \frac{C}{k}$) improves model performance compared to the natural, inverse square distribution ($P(\text{gcd} = k) = \frac{C}{k^2}$) . In this section, I experiment with three alternative distributions of outcomes:

- a "long-tail" log-uniform distribution: instead of sampling GCD between 1 and 100, they are sampled between 1 and 200,
- an inverse square root distribution of outcomes: $P(\text{gcd} = k) = \frac{C}{\sqrt{k}}$,
- an inverse power 1.5 distribution: $P(\text{gcd} = k) = \frac{C}{k\sqrt{k}}$.

Table 13: **Correct GCD for different outcome distribution scaling laws.** Best of 3 models, trained for 1000-1300 epochs. Log-uniform operands.

| Base | Outcome distribution scaling law | | | | |
|---|---|---|---|---|---|
| | $\frac{1}{k^2}$ | $\frac{1}{k\sqrt{k}}$ | $\frac{1}{k}, k \leq 100$ | $\frac{1}{k}, k \leq 200$ | $\frac{1}{\sqrt{k}}$ |
| 1000 | 71 | 71 | 91 | 90 | 91 |
| 1024 | 71 | 72 | 90 | 85 | 91 |
| 2017 | 63 | 64 | 88 | 87 | 88 |
| 2021 | 66 | 71 | 89 | 87 | 92 |
| 2023 | 65 | 67 | 88 | 85 | 90 |
| 2025 | 70 | 71 | 91 | 88 | 92 |
| 2187 | 66 | 70 | 91 | 86 | 91 |
| 2197 | 68 | 65 | 90 | 85 | 91 |
| 2209 | 65 | 68 | 87 | 85 | 90 |
| 2401 | 73 | 69 | 89 | 85 | 92 |
| 2744 | 72 | 72 | 91 | 88 | 89 |
| 3125 | 65 | 67 | 91 | 87 | 92 |
| 3375 | 67 | 68 | 91 | 87 | 92 |
| 4000 | 66 | 60 | 90 | 85 | 90 |
| 4913 | 57 | 60 | 88 | 90 | 92 |
| 5000 | 64 | 65 | 89 | 90 | 91 |
| 10000 | 56 | 55 | 88 | 90 | 91 |

Table 13 presents results for 17 bases between 1000 and 10000, for models trained with log-uniform operands and five distributions of outcomes. As observed in section 5, a log-uniform distribution of outcomes achieves better performances than the natural (inverse square) distribution. The inverse power 1.5 distribution of outcomes only brings marginal improvement over the natural distribution. With log-uniform outcomes, sampling GCD up to 200 instead of 100 has a negative impact on the number of correct GCD, except for the largest bases. On the other hand, training from an inverse square root distribution of outcomes improves performance for all bases. For 6 bases, 92 GCD under 100 are predicted correctly.

## D.2    LEARNING WITH SMALLER BATCHES

A common advice, when training transformers on natural language processing tasks, is to use the largest possible batches (i.e. as many as will fit in GPU memory). Large batches have two advantages, they avoid extreme gradients by averaging them over many samples, and they accelerate training by reducing the number of optimization steps. All models in this paper were trained with batches of 256 examples. In this section, I experiment with batches of 64, training models with log-uniform operands (and various outcome distributions) for about 800 epochs.

Table 14 compares models with batches of 64 to batches of 256, trained for a week (about 800 epochs for batch 64, 1300 for batch 256), on 11 different bases. For the same training time, batch size seems to have little impact on performance. This suggests that models could be trained on machines with less GPU memory, at no penalty.

Table 14: **Correct GCD for different batch sizes.** Best of 3 models, log-uniform operands. Models with batch size 64 are trained for 800 epochs, models with batch size 256 for 1300 epochs.

| Base | Inverse square outcomes | | Log-uniform outcomes | |
|---|---|---|---|---|
| | batch size 64 | batch size 256 | batch size 64 | batch size 256 |
| 10 | 49 | 48 | 69 | 65 |
| 12 | 54 | 55 | 67 | 70 |
| 30 | 56 | 56 | 73 | 72 |
| 31 | 45 | 44 | 64 | 64 |
| 210 | 55 | 60 | 81 | 80 |
| 1000 | 70 | 71 | 91 | 91 |
| 2025 | 66 | 70 | 90 | 91 |
| 2401 | 68 | 73 | 90 | 89 |
| 2744 | 70 | 72 | 91 | 91 |
| 4000 | 67 | 66 | 90 | 90 |
| 10000 | 55 | 56 | 89 | 88 |

## D.3    UNIFORM OPERANDS AND OUTCOMES - BASE 1000

In this section, I provide detailed results for models using base 1000, and trained on uniform operands and outcomes. Learning curves (Figure 4) are similar to those for base 10 (Figure 3) during the first 200 epochs: loss curves are flat, accuracy varies wildly, and the number of correct GCD has the characteristic step-like shape observed throughout this paper. Grokking, characterized by steep drops in the loss and increases in the number of correct GCD, happens between epochs 200 and 400. Then, the accuracy and the number of correct GCD, rise steadily. After 800 epochs 95 (out of 100) GCD are correctly predicted.

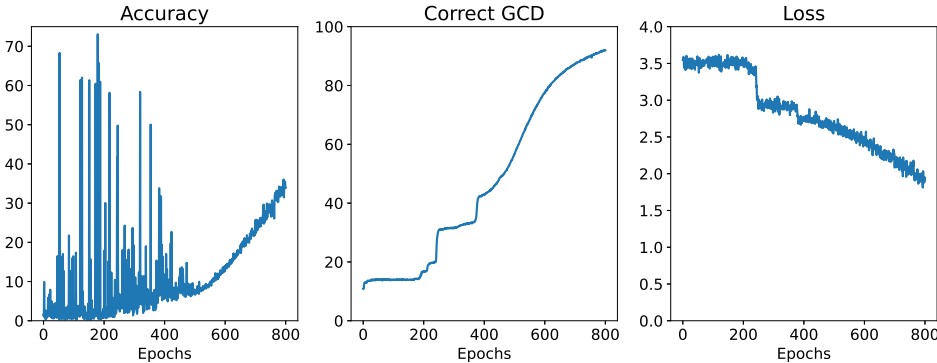

Figure 4: Learning curves for B=1000 - uniform operands and outcomes.

More precisely, by epoch 180, the model has learned to classify all examples into 14 sets: multiples of 1, 2, 4, 5, 8, 10, 16, 20, 25, 32, 40, 50, 80 and 100. At each epoch, the model selects one element in each class, which is its unique prediction for all pairs of integers with GCD in the class: the rules U1 to U3 are respected.

Grokking sets in around epoch 200, and by epoch 220, 5 new classes have been learned: multiples of 11 (11, 33, 77 and 99), 22, 44, 55 and 88, created by "splitting away" the multiples of 11 from the classes of multiples of 1, 2, 4, 5 and 8. Because of uniform outcomes, grokking does not happen in increasing order: 11 is learned before 3. By epoch 260, multiples of 3 are learned and the model predicts 31 different outcomes (splitting 12 classes, from 1 to 32). By epoch 400, multiples of 7 are learned, and 41 GCD are predicted.

During the grokking phase, a new phenomenon develops. As new primes are grokked and more classes are created, model predictions for each class become less deterministic. Instead of predicting a unique value for each class at each epoch, the model now "hesitates" between several values, and the frequency of the most common prediction goes down. By epoch 400, for the class of multiples of 1, the model makes 18 different predictions with frequencies ranging from 2% to 13% (Table 15).

Table 15: Base 1000 - epoch 400 - predicted values and frequencies.

| GCD 1 | | GCD 2 | | GCD 3 | | GCD 4 | | GCD 5 | |
|---|---|---|---|---|---|---|---|---|---|
| Pred. | % | Pred | % | Pred | % | Pred | % | Pred | % |
| 11 | 5 | 2 | 8 | 3 | 12 | 4 | 40 | 5 | 12 |
| 17 | 2 | 22 | 13 | 27 | 11 | 44 | 19 | 55 | 21 |
| 19 | 2 | 34 | 18 | 33 | 7 | 68 | 5 | 85 | 31 |
| 23 | 3 | 38 | 12 | 51 | 9 | 76 | 24 | 95 | 36 |
| 29 | 5 | 46 | 10 | 57 | 12 | 92 | 12 | | |
| 31 | 7 | 58 | 10 | 69 | 22 | | | | |
| 37 | 5 | 62 | 10 | 81 | 7 | | | | |
| 41 | 13 | 74 | 4 | 87 | 7 | | | | |
| 43 | 8 | 82 | 8 | 93 | 8 | | | | |
| 59 | 1 | 86 | 6 | 99 | 2 | | | | |
| 61 | 2 | | | | | | | | |
| 67 | 2 | | | | | | | | |
| 71 | 4 | | | | | | | | |
| 73 | 3 | | | | | | | | |
| 79 | 9 | | | | | | | | |
| 83 | 13 | | | | | | | | |
| 89 | 7 | | | | | | | | |
| 97 | 7 | | | | | | | | |

At this point, model predictions are neither deterministic nor interpretable, and the three rules are no longer respected. Classes have as many predictions as there are elements, and the model begins learning individual GCD, beginning with the largest ones (i.e. the smallest classes). By epoch 740, 95 of the 100 first GCD are correctly predicted, the worst performance being achieved on the smallest values (GCD 1, 2 and 3, correctly predicted 43, 74 and 85% of the time).

### D.4 UNIFORM OUTCOMES - LARGER BASES

In these experiments, models are trained for 1200 epochs, from uniform and log-uniform operands, and uniform outcomes. As previously, large bases achieve the best results. Models trained on uniform operands also seem to perform better.

Table 16: **Correct GCD with uniform outcomes.** Best of 3 models, trained for 1200 epochs.

| Base | Uniform operands | Log-uniform | Base | Uniform operands | Log-uniform |
|---|---|---|---|---|---|
| 1000 | 71 | 94 | 2401 | 98 | 90 |
| 2017 | 98 | 87 | 2744 | 96 | 93 |
| 2023 | 99 | 90 | 3375 | 98 | 92 |
| 2187 | 99 | 94 | 4000 | 98 | 94 |
| 2209 | 57 | 90 | 4913 | 99 | 93 |
| 2310 | 96 | 92 | 10000 | 99 | 95 |

### D.5 EXPERIMENTS WITH DIFFERENT ARCHITECTURES

In this section, I experiment with two popular recurrent architectures: long short-term memories (LSTM) (Hochreiter & Schmidhuber, 1997), and gated recurrent units (GRU) (Cho et al., 2014). I train models with 1024 and 2048 dimensions, and four layers, on uniform operands, log-uniform operands and log-uniform operands and outcomes, for 10 different bases: 10, 30, 31, 210, 420, 1000, 2021, 2023, 2025 and 2401.

By and large, the reuslts of my experiments with transformers extend to other recurrent networks. After 500 epochs, models trained on uniform operands (table 17 achieve performances similar to those obtained in sections 3 and 4. Composite bases like 210 and 420 achieve the best results (35 and 38 GCD), and large bases allow for grokking small primes. There is no clear advantage of LSTM over GRU, or 2048 over 124 dimensions. Models trained on log-uniform operands (table 18) and outcomes (table 19) perform better, but results are lower (after comparable training time) than with trasnformers.

Table 17: **Correct GCD with uniform operands.** Best of 3 models, trained for 500 epochs.

| Base | LSTM | | GRU | |
|---|---|---|---|---|
| | 1024 dim. | 2048 dim. | 1024 dim. | 2048 dim. |
| 10 | 15 | 15 | 15 | 15 |
| 30 | 32 | 30 | 32 | 30 |
| 31 | 2 | 2 | 2 | 2 |
| 210 | 35 | **35** | **35** | 35 |
| 420 | **38** | 34 | 34 | **38** |
| 1000 | 14 | 22 | 14 | 14 |
| 2021 | 8 | 7 | 8 | 10 |
| 2023 | 6 | 11 | 11 | 11 |
| 2025 | 24 | 24 | 10 | 18 |
| 2401 | 8 | 6 | 10 | 10 |

Table 18: **Correct GCD with log-uniform operands.** Best of 3 models, trained for 500 epochs.

| Base | LSTM | | GRU | |
|---|---|---|---|---|
| | 1024 dim. | 2048 dim. | 1024 dim. | 2048 dim. |
| 10 | 30 | 33 | 45 | 36 |
| 30 | 38 | 40 | 40 | 40 |
| 31 | 29 | 32 | 22 | 20 |
| 210 | 47 | 46 | 44 | 46 |
| 420 | 50 | 45 | 44 | 43 |
| 1000 | **53** | 51 | **46** | **47** |
| 2021 | 44 | 40 | 36 | 35 |
| 2023 | 46 | 48 | 38 | 39 |
| 2025 | 52 | **52** | 40 | 46 |
| 2401 | 47 | 41 | 35 | 33 |

Table 19: **Correct GCD with log-uniform operands and outcomes.** Best of 3 models, trained for 450 epochs.

| Base | LSTM | | GRU | |
|---|---|---|---|---|
| | 1024 dim. | 2048 dim. | 1024 dim. | 2048 dim. |
| 10 | 53 | 54 | 38 | 53 |
| 30 | 40 | 58 | 36 | 40 |
| 31 | 44 | 58 | 37 | 44 |
| 210 | 61 | 74 | 53 | 61 |
| 420 | 64 | 74 | 53 | 64 |
| 1000 | **69** | 73 | **62** | **69** |
| 2021 | 60 | **76** | 54 | 60 |
| 2023 | 65 | 73 | 53 | 65 |
| 2025 | **69** | 74 | 60 | **69** |
| 2401 | 55 | 73 | 50 | 55 |

# E ADDITIONAL RESULTS

## E.1 GROKKING

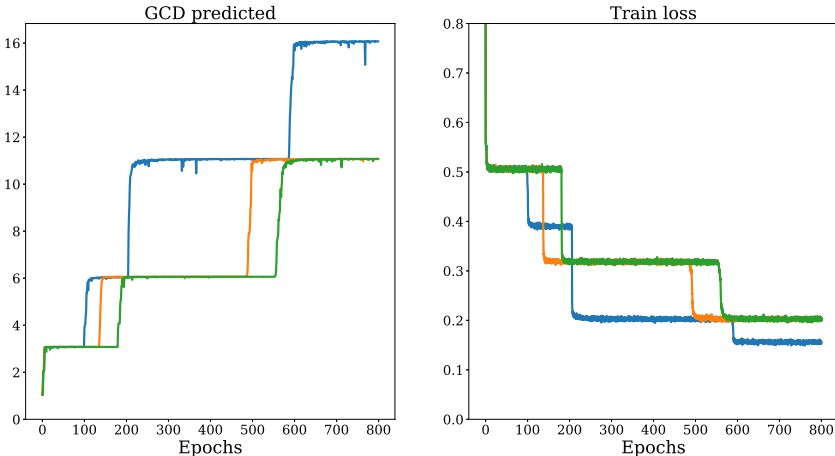

Figure 5: **Learning curves for base B=2023.** 3 different model initializations.

Table 20: **Model predictions.** $B = 1000$, after 220 epochs. 32 is being learned.

| GCD | Prediction | GCD | Prediction | GCD | Prediction | GCD | Prediction | GCD | Prediction |
|-----|-----------|-----|-----------|-----|-----------|-----|-----------|-----|-----------|
| 1 | 1 | 11 | 1 | 21 | 3 | 31 | 1 | 41 | 1 |
| 2 | 2 | 12 | 12 | 22 | 2 | 32 | 16/ 32 | 42 | 6 |
| 3 | 3 | 13 | 1 | 23 | 1 | 33 | 3 | 43 | 1 |
| 4 | 4 | 14 | 2 | 24 | 24 | 34 | 2 | 44 | 4 |
| 5 | 5 | 15 | 15 | 25 | 25 | 35 | 5 | 45 | 15 |
| 6 | 6 | 16 | 16 | 26 | 2 | 36 | 12 | 46 | 2 |
| 7 | 1 | 17 | 1 | 27 | 3 | 37 | 1 | 47 | 1 |
| 8 | 8 | 18 | 6 | 28 | 4 | 38 | 2 | 48 | 48 |
| 9 | 3 | 19 | 1 | 29 | 1 | 39 | 3 | 49 | 1 |
| 10 | 10 | 20 | 20 | 30 | 30 | 40 | 40 | 50 | 50 |

Table 21: **Predicted GCD, natural test distribution, and 5% uniform GCD .** Best model of 3. .

| Base | Natural distribution | | 5% uniform GCD | |
| | Correct GCD | Epochs | Correct GCD | Epochs |
|------|-------------|--------|-------------|--------|
| 625 | 6 | 650 | 3 | 10 |
| 1000 | 22 | 250 | 15 | 560 |
| 2017 | 4 | 450 | 1 | 0 |
| 2021 | 10 | 550 | 10 | 600 |
| 2023 | 16 | 600 | 11 | 800 |
| 2025 | 28 | 850 | 18 | 225 |
| 2187 | 20 | 750 | 12 | 750 |
| 2197 | 11 | 800 | 11 | 775 |
| 2209 | 8 | 850 | 6 | 575 |
| 2401 | 14 | 700 | 14 | 630 |
| 2744 | 29 | 1400 | 21 | 650 |
| 3125 | 16 | 550 | 11 | 500 |
| 3375 | 23 | 400 | 23 | 475 |
| 4000 | 25 | 650 | 25 | 600 |
| 4913 | 17 | 950 | 7 | 575 |
| 5000 | 28 | 900 | 24 | 675 |
| 10000 | 22 | 250 | 22 | 300 |

### E.2 LEARNING CURVES - LOG-UNIFORM OPERANDS

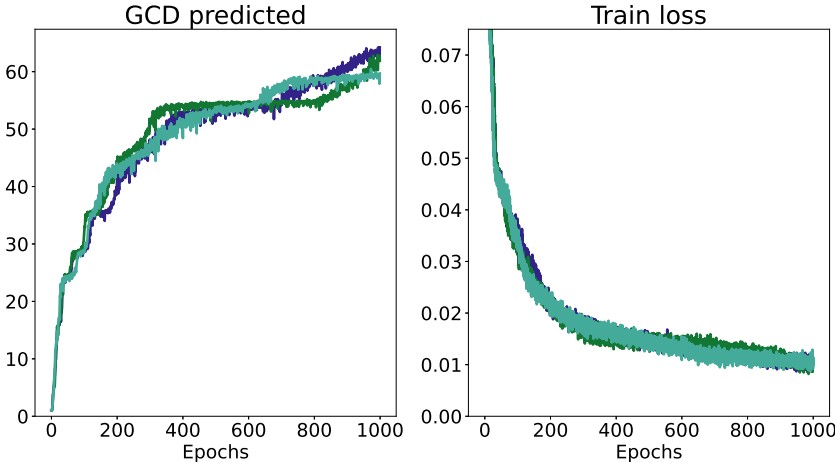

Figure 6: **Learning curves for base B=2023. Log-uniform operands, natural outcomes.** 3 different model initializations.

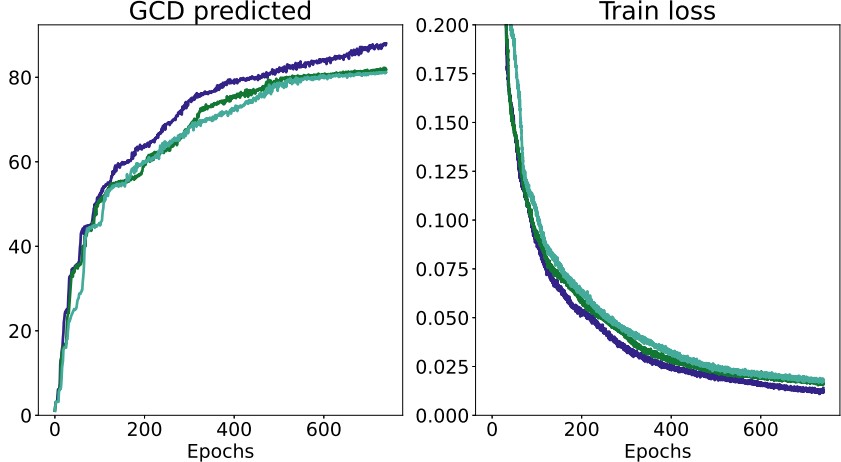

Figure 7: **Learning curves for base B=2023. Log-uniform operands, log-uniform outcomes.** 3 different model initializations.

### E.3 DETAILED MODEL PREDICTIONS - BASE EXPERIMENTS

Table 22: Predicted values for gcd 1 to 63.

| Base GCD | 2 Prediction | % | 4 Pred. | % | 10 Pred. | % | 30 Pred. | % | 31 Pred. | % | 420 Pred. | % |
|---|---|---|---|---|---|---|---|---|---|---|---|---|
| 1 | **1** | 100 | **1** | 100 | **1** | 100 | **1** | 100 | **1** | 100 | **1** | 100 |
| 2 | **2** | 100 | **2** | 100 | **2** | 100 | **2** | 100 | 1 | 100 | **2** | 100 |
| 3 | 1 | 100 | 1 | 100 | 1 | 100 | **3** | 100 | 1 | 100 | **3** | 100 |
| 4 | **4** | 100 | **4** | 100 | **4** | 100 | **4** | 100 | 1 | 100 | **4** | 100 |
| 5 | 1 | 100 | 1 | 100 | **5** | 100 | **5** | 100 | 1 | 100 | **5** | 100 |
| 6 | 2 | 100 | 2 | 100 | 2 | 100 | **6** | 100 | 1 | 100 | **6** | 99.6 |
| 7 | 1 | 100 | 1 | 100 | 1 | 100 | 1 | 100 | 1 | 100 | **7** | 100 |
| 8 | **8** | 100 | **8** | 100 | **8** | 100 | **8** | 100 | 1 | 100 | **8** | 100 |
| 9 | 1 | 100 | 1 | 100 | 1 | 100 | **9** | 100 | 1 | 100 | **9** | 100 |
| 10 | 2 | 100 | 2 | 100 | 10 | 100 | 10 | 100 | 1 | 100 | 10 | 100 |
| 11 | 1 | 100 | 1 | 100 | 1 | 100 | 1 | 100 | 1 | 100 | 1 | 100 |
| 12 | 4 | 100 | 4 | 100 | 4 | 100 | 12 | 100 | 1 | 100 | 12 | 99.8 |
| 13 | 1 | 100 | 1 | 100 | 1 | 100 | 1 | 100 | 1 | 100 | 1 | 100 |
| 14 | 2 | 100 | 2 | 100 | 2 | 100 | 2 | 100 | 1 | 100 | 14 | 100 |
| 15 | 1 | 100 | 1 | 100 | 5 | 100 | 15 | 100 | 1 | 100 | 15 | 99.4 |
| 16 | 16 | 100 | 16 | 100 | 16 | 99.7 | 8 | 100 | 1 | 100 | 16 | 100 |
| 17 | 1 | 100 | 1 | 100 | 1 | 100 | 1 | 100 | 1 | 100 | 1 | 100 |
| 18 | 2 | 100 | 2 | 100 | 2 | 100 | 18 | 100 | 1 | 100 | 18 | 100 |
| 19 | 1 | 100 | 1 | 100 | 1 | 100 | 1 | 100 | 1 | 100 | 1 | 100 |
| 20 | 4 | 100 | 4 | 100 | 20 | 100 | 20 | 100 | 1 | 100 | 20 | 100 |
| 21 | 1 | 100 | 1 | 100 | 1 | 100 | 3 | 100 | 1 | 100 | 21 | 100 |
| 22 | 2 | 100 | 2 | 100 | 2 | 100 | 2 | 100 | 1 | 100 | 2 | 100 |
| 23 | 1 | 100 | 1 | 100 | 1 | 100 | 1 | 100 | 1 | 100 | 1 | 100 |
| 24 | 8 | 100 | 8 | 100 | 8 | 100 | 24 | 100 | 1 | 100 | 24 | 100 |
| 25 | 1 | 100 | 1 | 100 | 25 | 100 | 25 | 99 | 1 | 100 | 25 | 99.9 |
| 26 | 2 | 100 | 2 | 100 | 2 | 100 | 2 | 100 | 1 | 100 | 2 | 100 |
| 27 | 1 | 100 | 1 | 100 | 1 | 100 | 9 | 100 | 1 | 100 | 9 | 100 |
| 28 | 4 | 100 | 4 | 100 | 4 | 100 | 4 | 100 | 1 | 100 | 28 | 100 |
| 29 | 1 | 100 | 1 | 100 | 1 | 100 | 1 | 100 | 1 | 100 | 1 | 100 |
| 30 | 2 | 100 | 2 | 100 | 10 | 100 | 30 | 100 | 1 | 100 | 30 | 99.6 |
| 31 | 1 | 100 | 1 | 100 | 1 | 100 | 1 | 100 | 31 | 100 | 1 | 100 |
| 32 | 32 | 99.9 | 32 | 98.7 | 16 | 99.9 | 8 | 100 | 1 | 100 | 16 | 100 |
| 33 | 1 | 100 | 1 | 100 | 1 | 100 | 3 | 100 | 1 | 100 | 3 | 100 |
| 34 | 2 | 100 | 2 | 100 | 2 | 100 | 2 | 100 | 1 | 100 | 2 | 100 |
| 35 | 1 | 100 | 1 | 100 | 5 | 100 | 5 | 100 | 1 | 100 | 35 | 100 |
| 36 | 4 | 100 | 4 | 100 | 4 | 100 | 36 | 100 | 1 | 100 | 36 | 100 |
| 37 | 1 | 100 | 1 | 100 | 1 | 100 | 1 | 100 | 1 | 100 | 1 | 100 |
| 38 | 2 | 100 | 2 | 100 | 2 | 100 | 2 | 100 | 1 | 100 | 2 | 100 |
| 39 | 1 | 100 | 1 | 100 | 1 | 100 | 3 | 100 | 1 | 100 | 3 | 99.9 |
| 40 | 8 | 99.9 | 8 | 100 | 40 | 99.9 | 40 | 100 | 1 | 100 | 40 | 99.9 |
| 41 | 1 | 100 | 1 | 100 | 1 | 100 | 1 | 100 | 1 | 100 | 1 | 100 |
| 42 | 2 | 100 | 2 | 100 | 2 | 100 | 6 | 99.9 | 1 | 100 | 42 | 100 |
| 43 | 1 | 100 | 1 | 100 | 1 | 100 | 1 | 100 | 1 | 100 | 1 | 100 |
| 44 | 4 | 100 | 4 | 100 | 4 | 100 | 4 | 100 | 1 | 100 | 4 | 100 |
| 45 | 1 | 100 | 1 | 100 | 5 | 100 | 45 | 100 | 1 | 100 | 45 | 99.8 |
| 46 | 2 | 100 | 2 | 100 | 2 | 100 | 2 | 100 | 1 | 100 | 2 | 100 |
| 47 | 1 | 100 | 1 | 100 | 1 | 100 | 1 | 100 | 1 | 100 | 1 | 100 |
| 48 | 16 | 100 | 16 | 100 | 16 | 99.9 | 24 | 100 | 1 | 100 | 48 | 99.9 |
| 49 | 1 | 100 | 1 | 100 | 1 | 100 | 1 | 100 | 1 | 100 | 7 | 100 |
| 50 | 2 | 100 | 2 | 100 | 50 | 100 | 50 | 100 | 1 | 100 | 50 | 99.6 |
| 51 | 1 | 100 | 1 | 100 | 1 | 100 | 3 | 100 | 1 | 100 | 3 | 99.8 |
| 52 | 4 | 100 | 4 | 100 | 4 | 100 | 4 | 100 | 1 | 100 | 4 | 100 |
| 53 | 1 | 100 | 1 | 100 | 1 | 100 | 1 | 100 | 1 | 100 | 1 | 100 |
| 54 | 2 | 100 | 2 | 100 | 2 | 100 | 18 | 99.9 | 1 | 100 | 18 | 100 |
| 55 | 1 | 100 | 1 | 100 | 5 | 100 | 5 | 100 | 1 | 100 | 5 | 100 |
| 56 | 8 | 100 | 8 | 100 | 8 | 99.9 | 8 | 100 | 1 | 100 | 56 | 100 |
| 57 | 1 | 100 | 1 | 100 | 1 | 100 | 3 | 100 | 1 | 100 | 3 | 99.9 |
| 58 | 2 | 100 | 2 | 100 | 2 | 100 | 2 | 100 | 1 | 100 | 2 | 100 |
| 59 | 1 | 100 | 1 | 100 | 1 | 100 | 1 | 100 | 1 | 100 | 1 | 100 |
| 60 | 4 | 100 | 4 | 100 | 20 | 100 | 60 | 100 | 1 | 100 | 60 | 99.7 |
| 61 | 1 | 100 | 1 | 100 | 1 | 100 | 1 | 100 | 1 | 100 | 1 | 100 |
| 62 | 2 | 100 | 2 | 100 | 2 | 100 | 2 | 100 | 31 | 100 | 2 | 100 |
| 63 | 1 | 100 | 1 | 100 | 1 | 100 | 9 | 100 | 1 | 100 | 63 | 100 |

Table 23: Predicted values for gcd 64 to 100.

| Base GCD | 2 Prediction | % | 4 Pred. | % | 10 Pred. | % | 30 Pred. | % | 31 Pred. | % | 420 Pred. | % |
|---|---|---|---|---|---|---|---|---|---|---|---|---|
| 64 | 64 | 98.9 | 64 | 99.2 | 16 | 99.8 | 8 | 100 | 1 | 100 | 16 | 100 |
| 65 | 1 | 100 | 1 | 100 | 5 | 100 | 5 | 100 | 1 | 100 | 5 | 100 |
| 66 | 2 | 100 | 2 | 100 | 2 | 100 | 6 | 100 | 1 | 100 | 6 | 100 |
| 67 | 1 | 100 | 1 | 100 | 1 | 100 | 1 | 100 | 1 | 100 | 1 | 100 |
| 68 | 4 | 100 | 4 | 100 | 4 | 100 | 4 | 100 | 1 | 100 | 4 | 100 |
| 69 | 1 | 100 | 1 | 100 | 1 | 100 | 3 | 100 | 1 | 100 | 3 | 100 |
| 70 | 2 | 100 | 2 | 100 | 10 | 100 | 10 | 100 | 1 | 100 | 70 | 100 |
| 71 | 1 | 100 | 1 | 100 | 1 | 100 | 1 | 100 | 1 | 100 | 1 | 100 |
| 72 | 8 | 100 | 8 | 100 | 8 | 100 | 72 | 100 | 1 | 100 | 72 | 100 |
| 73 | 1 | 100 | 1 | 100 | 1 | 100 | 1 | 100 | 1 | 100 | 1 | 100 |
| 74 | 2 | 100 | 2 | 100 | 2 | 100 | 2 | 100 | 1 | 100 | 2 | 100 |
| 75 | 1 | 100 | 1 | 100 | 25 | 100 | 75 | 100 | 1 | 100 | 75 | 99.4 |
| 76 | 4 | 100 | 4 | 100 | 4 | 100 | 4 | 100 | 1 | 100 | 4 | 100 |
| 77 | 1 | 100 | 1 | 100 | 1 | 100 | 1 | 100 | 1 | 100 | 7 | 100 |
| 78 | 2 | 100 | 2 | 100 | 2 | 100 | 6 | 100 | 1 | 100 | 6 | 100 |
| 79 | 1 | 100 | 1 | 100 | 1 | 100 | 1 | 100 | 1 | 100 | 1 | 100 |
| 80 | 16 | 100 | 16 | 100 | 80 | 99.9 | 40 | 100 | 1 | 100 | 80 | 100 |
| 81 | 1 | 100 | 1 | 100 | 1 | 100 | 9 | 100 | 1 | 100 | 9 | 99.8 |
| 82 | 2 | 100 | 2 | 100 | 2 | 100 | 2 | 100 | 1 | 100 | 2 | 100 |
| 83 | 1 | 100 | 1 | 100 | 1 | 100 | 1 | 100 | 1 | 100 | 1 | 100 |
| 84 | 4 | 100 | 4 | 100 | 4 | 100 | 12 | 100 | 1 | 100 | 84 | 100 |
| 85 | 1 | 100 | 1 | 100 | 5 | 100 | 5 | 100 | 1 | 100 | 5 | 100 |
| 86 | 2 | 100 | 2 | 100 | 2 | 100 | 2 | 100 | 1 | 100 | 2 | 100 |
| 87 | 1 | 100 | 1 | 100 | 1 | 100 | 3 | 100 | 1 | 100 | 3 | 99.8 |
| 88 | 8 | 100 | 8 | 100 | 8 | 100 | 8 | 100 | 1 | 100 | 8 | 100 |
| 89 | 1 | 100 | 1 | 100 | 1 | 100 | 1 | 100 | 1 | 100 | 1 | 100 |
| 90 | 2 | 100 | 2 | 100 | 10 | 100 | 90 | 100 | 1 | 100 | 90 | 99.9 |
| 91 | 1 | 100 | 1 | 100 | 1 | 100 | 1 | 100 | 1 | 100 | 7 | 100 |
| 92 | 4 | 99.9 | 4 | 100 | 4 | 100 | 4 | 100 | 1 | 100 | 4 | 100 |
| 93 | 1 | 100 | 1 | 100 | 1 | 100 | 3 | 100 | 31 | 99.9 | 3 | 99.8 |
| 94 | 2 | 100 | 2 | 100 | 2 | 100 | 2 | 100 | 1 | 100 | 2 | 100 |
| 95 | 1 | 100 | 1 | 100 | 5 | 100 | 5 | 100 | 1 | 100 | 5 | 100 |
| 96 | 32 | 100 | 32 | 99.5 | 16 | 99.8 | 24 | 100 | 1 | 100 | 48 | 99.9 |
| 97 | 1 | 100 | 1 | 100 | 1 | 100 | 1 | 100 | 1 | 100 | 1 | 100 |
| 98 | 2 | 100 | 2 | 100 | 2 | 100 | 2 | 100 | 1 | 100 | 14 | 100 |
| 99 | 1 | 100 | 1 | 100 | 1 | 100 | 9 | 100 | 1 | 100 | 9 | 99.8 |
| 100 | 4 | 100 | 4 | 100 | 100 | 100 | 100 | 100 | 1 | 100 | 100 | 99.6 |

