# OpenReview forum: "Learning the greatest common divisor: explaining transformer predictions"
_ICLR.cc/2024/Conference — ICLR 2024 spotlight_

### Official Review · Reviewer_zrzn · 2023-10-31

**Soundness:** 3 good
**Presentation:** 2 fair
**Contribution:** 2 fair
**Rating:** 5
**Confidence:** 4

**Summary:**

The paper presents an exploration of using small transformers to calculate the greatest common divisor (GCD) of two positive integers. A notable aspect is that the predictions made by these models are explainable. The authors focus on how models learn a list of divisors and predict the largest element that divides both inputs. They also investigate the impact of different training distributions on model performance, including uniform operands, log-uniform operands, and balanced distributions of GCDs. They also investigate with different inputs representations in different bases.

**Strengths:**

- Studying learning GCD through transformers is a novel problem.
- This work raises some observations about the deterministic behavior of the model in learning GCD, and the simple shortcut algorithm it learns.
- A lot of experiments/examples supporting the claims are provided.

**Weaknesses:**

My main concern is about the significance of studying computing GCD through transformers. The observations are all limited to this particular task and it is not clear how broadly applicable they are. Or, if there is anything to learn from these observations and apply it elsewhere.

Moreover, the observations are also not robust. The authors point out that the algorithm learned is deterministic but then this observations breaks down when the input distribution is changed. This non-robust behavior of the observations weaken them a lot in my opinion. For example, explainability breaks down with the change in input distribution.

**Questions:**

- Did the authors try to experiment with Chain-of-Thought and see if providing the explanation helps with the accuracy?
- Can the authors please explain why does the model only learn divisors of the base B? And, if there is a way to promote learning of other divisors?
- The authors mention that some non-divisible small primes are learned very late in the training stage. Did the authors see that if they continue training further, eventually other primes will be learned?
- The authors mention that explainability of the model breaks down with the change in the input distribution. Do the authors have any thoughts on why this is the case?
- Do few shot prompts improve the performance?
- The authors mention they observe grokking phenomenon. Can the authors please provide more related work on grokking and transformers. And, if this is the first work observing grokking for transformers.

---

> ### Author Response · Authors · 2023-11-19
> **reply to reviewer zrzn**
>
> Thank you for your comments, questions and suggestions. We will integrate our replies in the revised version (in a few days).
>
> *My main concern is about the significance of studying computing GCD through transformers.*
>
> The broader context of our research can be summarized as follows :
>
> * This is a first step towards using mathematical tasks to understand and explain deep learning models, by investigating their predictions on selected tasks. This extends current approaches on explainability which mostly focus on mechanical interpretability - i.e. looking at the model weights and internal calculations. In this paper, we demonstrate the feasibility of such an approach on a non-trivial mathematical task. By leveraging our theoretical knowledge of the underlying problem, we can design train and test sets, and get a clear picture of how the model learns to solve the problem. We believe that running similar analyses on a larger set of mathematical problems would provide a lot of insight about the inner workings of deep learning models (not just transformers).
> * While tasks of pure mathematics are not the main focus of LLM practitioners, they play a central role for the development of foundational models for science: large language models pre-trained on mathematical equations, instead of natural language. Our results show how transformers can be trained to perform exact calculations involving integer divisibility, a central task in integer arithmetic and number theory.
> * Our results on training distributions, i.e. the fact that some distributions allow for faster learning and better out-of-distribution performance, may apply to other arithmetic tasks. In particular, log-uniform operands and outcomes could be used when fine tuning LLM, or training foundational models for physics.
>
> *Moreover, the observations are also not robust. The authors point out that the algorithm learned is deterministic but then this observations breaks down when the input distribution is changed.*
>
> Our observations on determinism and explainability, and in particular the three rules for explainability G1, G2, G3 (section 4), hold for uniform and log-uniform operands, and natural (inverse square) and log-uniform outcomes, together with the additional distributions discussed in appendix D1. They are also valid for other deep learning architectures (see our reply to reviewer 21KR). We believe this makes a strong case for their robustness.
>
> In fact, explainability only breaks (partially) when training on uniformly distributed GCD (section 6). The G rules, then get replaced by the weaker U rules, which break again (but temporarily) as the model learns the very last GCD (appendix D3, and figure 15). We believe this limiting case actually sheds light on what the model is learning. The non-uniform distribution of outcomes is an inductive bias about the problem that the model learns to exploit (see also our reply to your question below).
>
> *Can the authors please explain why does the model only learn divisors of the base B? And, if there is a way to promote learning of other divisors?*
>
> The divisors of B are learned because divisibility by these numbers can be checked by looking at the last digits of the integer representation in base B. For instance, in base 10, a number is divisible by 2 iff its representation ends with a 0, 2, 4, 6  or 8. It is divisible by 5 iff it ends in 5 or 0, and by 20 iff it ends in 00, 20, 40, 60 or 80. This means that the model can check whether both its operands are divisible by 20, by looking at their 2 last digits (the attention mechanism allows the transformer to point at specific digits in the sequences) and comparing them to a set of memorized values (the FFN layers in the transformer allow for such memorization). For other bases, such simple rules do not exist: all digits in the number representation must be taken into account when checking for divisibility. We believe this explains our results from section 3.
>
> In section 4, we observe that other divisors: small primes and their multiples, can be learned (i.e. grokked) when the model is trained long enough. How this phenomenon happens is not clear, but it seems to happen for all large bases, and to be accelerated by training on log-uniform distributions of operands and outcomes (section 5). In our best results, all products of primes up to 53 are learned. Observations from appendix D1 suggest that these results can be improved. Our experiments in section 6 indicate that an imbalance distribution of outcomes is required for this phenomenon to take place.

---

> ### Author Response · Authors · 2023-11-19
> **reply to reviewer zrzn (2/2)**
>
> *The authors mention that some non-divisible small primes are learned very late in the training stage. Did the authors see that if they continue training further, eventually other primes will be learned?*
>
> Yes. Our initial experiments with grokking, for B=1000, shows that whereas all divisors of B were learned after 84 epochs, it took 220 epochs to learn 3 and its multiples. The rightmost column in Table 4 shows, for models trained up to 1300 epochs, the number of epochs needed to learn each small divisor. For instance, for B=2023, 3 is learned after 101 epochs, 2 after 205 and 4 after 599. For B=2744, 3 is learned after 543 epochs, and 5 after 1315. We believe that longer training would help (but 1300 epochs takes over 2 weeks on the machines we used). This is suggested by our experiments with log-uniform operands, which accelerate grokking, and allow to learn all primes up to 23 with a “natural” distribution of outcomes, and 53 with a log-uniform distribution of outcomes.
>
> *The authors mention that explainability of the model breaks down with the change in the input distribution. Do the authors have any thoughts on why this is the case?*
>
> Explainability does not break down when the input distribution, i.e. the distribution of operands, changes (in section 5). The three rules G1, G2 and G3 from section 4 still hold for models trained on log-uniform operands, and even log-uniform outcomes. The only difference, mentioned in the last paragraph of section 5 (bottom of page 6), is that the transitions between different phases – the moment when the model is learning a new divisor –  now span several epochs, during which model predictions are split between the old and the new value. Out of these transient phases, model predictions remain deterministic and explainable.
>
> Explainability (partly) breaks when models are trained on uniform outcomes. This is described in section 6. Models trained on uniform outcomes still learn to cluster their input pairs into classes which correspond to products of divisors of the base. All elements in the class are usually predicted the same, and new classes of multiples of new primes are learned as training proceeds (see the correct GCD in figure 3). Yet, the model prediction for each class varies from epoch to epoch (Table 8): multiples of 4 are predicted sometimes as 4, sometimes as 44, but at a given time in training, all multiples are predicted the same.
>
> We believe this is an effect of the uniform distribution of outcomes: in previous experiments, the model predicted the most common instance of each class, which turned to be its smallest member. As a result, all multiples of 4 would be predicted as 4, and 1 would be the default prediction for all “non predictable” GCD. With uniform outcomes, the most common instance randomly varies over time, and this accounts for the fluctuations observed in table 8. In other words, the distribution of outcomes  in the training set provides the model with an inductive bias about divisibility, which disappears when outcomes are uniformly sampled.
>
> *Did the authors try to experiment with Chain-of-Thought and see if providing the explanation helps with the accuracy?*
>
> Chain-of-thought does not apply to this setting, as it requires a decoder-only architecture, and we use encoder-decoder. Besides, it would demand that we know in advance the algorithm that the model is going to use. In the case of GCD, providing the model with steps from the “human algorithm” (i.e. Euclid’s algorithm) would certainly not help the model learn the sieving technique it is implementing.
>
> *The authors mention they observe grokking phenomenon. Can the authors please provide more related work on grokking and transformers. And, if this is the first work observing grokking for transformers.*
>
> The original paper on grokking (Power et al. 2022, arxiv 2201.02177 referenced in the paper), observed the phenomenon in a transformer learning modular arithmetic – i.e. arithmetic over a finite set. The follow-up papers by Liu et al. (https://arxiv.org/abs/2205.10343) and Gromov (https://arxiv.org/abs/2301.02679) mostly focus on fully connected networks (MLP). Nanda et al. (https://arxiv.org/abs/2301.05217) study grokking in one-layer transformers. All these works focus on arithmetic over finite sets: the model is tasked to learn, or complete, a finite “addition table”. To our knowledge, our work is the first observation of grokking on a different arithmetic task.

---

### Official Review · Reviewer_t8xf · 2023-10-31

**Soundness:** 3 good
**Presentation:** 3 good
**Contribution:** 3 good
**Rating:** 6
**Confidence:** 4

**Summary:**

This paper studies Transformers capabilities on learning the task of computing greatest common divisor (GCD) for two numbers. First the two integers of input and the integer of the output are encoded in base $B$. Then, the model is trained with 300,000 new samples at each epoch. The authors have tried different distributions for the input/operands (uniform and log-uniform) and also the output. They have observed that in the majority of their experiments, model's output is deterministic, meaning that if $\gcd(a,b)=k$, the output of the model is fixed depending on $k$. Further, for $k$'s that are a divisor of $B$ or small, the output is usually correct. Moreover, they show that by having a log-uniform distribution on the operands, the performance of the model is increased.

**Strengths:**

- The particular attention to the distribution of the operands is important, and it's shown that by emphasizing on small numbers, the performance of the model can be increased (resembling curriculum learning).
- Similarly, authors have considered the distribution of the output showing that large GCDs may be slow/hard to learn as they are rare.
- Experiments are rather extensive in several axes (e.g., number of bases, size of the models, batch-size, ...).

**Weaknesses:**

- The claim that Transformer predictions are fully explainable does not seem to be accurate once log-uniform operands are used.
- Although, the paper usually interprets the results of experiments, it does not put forward any explanation for such results (for example, defining and justifying the shortcuts Transformers may take).
- Some of the report rules might be consequences of other factors (and they may not be robust as a result). For example, the smaller primes are learned (or grokked) faster as they are more common in the output or really easier to learn? Although these problems can be clarified.
See the questions below for more details.

**Questions:**

- Q1. It's said that composite bases allow one to check for divisibility by seeing the rightmost digits. This is true, for example for powers of the base, one needs to count the number of zeros. However, the picture seems to be more complex for divisors of the base. For example, consider $B=210$. Any number ($<10^6$), can be expressed using 3 digits in this base. Now for checking divisibility by 8, one has to check the pattern of all 3 digits (and it's not as simple as being 0). On the other hand, checking divisibility for any other number also requires checking the 3 digits together. Is there anything that is making checking for 8 simpler? (Similarly, assume that $B=350$, could we expect that checking for $3, 6$ would be easier than $8$?)
- Q2. At the beginning of the paper there is the idea that divisors of $B$ are easily learned. Is this also the case when $B$ is for example, the product of two large primes, e.g., $B=2021=43\times 47$? Is there a possiblility that smaller primes are learned faster than $43, 47, 43^2, \ldots$ in this case?
- Q3. Also given the list intuition, why 420 is performing better than 210 in Table 2?
- Q4. Further, the list intuition does not seem to be applicable anymore in Table 6 (E.g., $2401=7^4$ is having the best performance). What could be the reason for this?


- Q5. In claim U2, one can see that 21 is wrongly classified into $C_4$. As a result I wonder, for other bases that are product of larger primes, e.g., again, $2021=43\times 47$, can this phenomenon still be observe? Or is having small primes such as 2 and 5 playing an important role here?
- Q6. In the line before table 5, we see that for $B=30$, $B-1, B+1$ are learned faster than other primes. I guess this might be due to the fact that $(x_3x_2x_1)_B \pmod{B-1} = x_3 \times B^2 + x_2 \times B + x_1 \pmod{B-1} =  x_3 + x_2  + x_1 \pmod{B-1}$ which may be easier to learn due to symmetry between $x_3, x_2, x_1$. Similar explanations can be given for any divisor of $B-1$ and $B+1$. If that's the case, this factor should also be considered when we see some primes are grokked.

## Minor questions/remarks
- Q7. What stopping criterion has been used? For example in Figure 4, the loss appears to be still decreasing. Maybe the combination of continued training and weight decay results in improved generalization as in grokking?
- Q8. The addition and multiplication of rationals mentioned in the intro and studied in the appendix require the output to be simplified. So I wonder, is there any particular difference or additional complexity comparing to the simplification task itself?
- Q9. What is the positional embedding used for the model?
- R1. My understanding is that the samples at each batch are freshly generated. I think it would be beneficial if this is communicated more clearly and earlier to the reader.
- R2. Table 5 is a bit misleading with the current format. For example, when we see than 9 is learned for $B=2017$ we have to guess that 3 was already learned. (Similar issue can be observed for other bases.)
- R3. For Section 5 experiments in page 6, it would helpful if training loss plot is provided.

---

> ### Author Response · Authors · 2023-11-19
> **Reply to reviewer t8xf**
>
> Thank you for your comments, questions and suggestions. We will integrate our replies in the revised version (in a few days).
>
> *The claim that Transformer predictions are fully explainable does not seem to be accurate once log-uniform operands are used.*
>
> The three rules for explainability (G1, G2, G3) are hold over a large set of training distributions : uniform and log-uniform operands, natural and log-uniform outcomes (together with the additional distributions from appendix D1). We will try to clarify the last sentence in the abstract.
>
> In fact, explainability only breaks (partially) when the training data uniformly samples GCD (section 6). The G rules, then get replaced by the weaker U rules, which break again (but temporarily) as the model learns the very last GCD (appendix D3, and figure 15). We see these last cases are limiting conditions. They suggest that an unbalanced distribution of outcomes is needed to explainability to happen.
>
> *Although, the paper usually interprets the results of experiments, it does not put forward any explanation for such results*
>
> A number of explanations are provided, but they are scattered through the paper. We will concentrate them in one subsection of the discussion. Summarizing:
>
> We can explain the shortcuts we believe our models use to test divisibility by products of divisors of the base. When the base is a prime number, the model counts the rightmost zeroes in both of its operands, takes the minimum $k$, and predicts $B^k$. When the base is composite (or a power of a prime), the model exploits the following property: if a factor $f$ divides $B^n$, ie $B^n = kf$, an integer $m$ is divisible by $f$ if and only if its $n$ last digits in base $B$ take one of the $k$ values : $0, f, 2f, … (k-1)f$. For instance, in base 10, a number is divisible by 20 iff it ends in 0, 20, 40, 60 or 80. We believe the model memorizes those lists when k and n are small. For small bases ($B<500$), models learn all divisors of $B^2$ this way.
>
> We also understand how the model leverages divisibility rules to predict GCD. This exploits the imbalance in the GCD distribution (i.e. the fact that, in the training set, small GCD are more common than large GCD). As training begins, the model predicts 1 (the most common value) for all input pairs $(a,b)$. When divisibility by $k$ is learned (via the shortcuts described above, or grokking), the model learns to distinguish all pairs of the form $(ka, kb)$, i.e. such that their GCD is a multiple of $k$. The model therefore predicts $k$ for all pairs of the form $(ka,kb)$, and $1$ for the others. Once a second divisor, $l$, is learned, both input classes (i.e. $(ka,kb)$ input predicted as $k$, and others, predicted as $1$), are split into four classes, predicted as $1$, $k$, $l$, and $kl$ (except if $l=k^2$, in which case we end up with 3 classes, predicted as $1$, $k$ and $k^2$). Every new prime divisor $p$, learned by the model, causes each previous class, predicted as $f$, to split into two new classes, predicted as $f$ and $pf$. This accounts for the steps observed in the training curves.
>
> Finally, we can explain how this process leads to a correct algorithm for predicting GCD, when outcomes are not uniformly distributed. Once the divisors of the base are learned, small primes are grokked in order (so far, we do not have an explanation for grokking), and new classes are created, that correspond to the products of the prime with previously learned factors. These classes are predicted as $pf$ ($p$ the prime, $f$ the previous factor), and contain all pairs $(pfa pfb)$, divisible by $pf$. Pairs with GCD $pf$ are now correctly predicted, and multiples of $pf$ will be learned once all their prime power divisors are learned or grokked.
>
> The only part in the process that cannot be explained is the grokking process. We leave it to future work.
> We will add these explanations to the discussion section.
>
> *the smaller primes are learned (or grokked) faster as they are more common in the output or really easier to learn?*
>
> The experiments from section 6 shed light on this. Non-divisors are grokked in order, and each class is predicted as its smallest element (i.e. its largest common divisor) so long the distribution of outcomes in the training set in imbalanced.  This suggests that small primes are learned first, because small GCD are more common in the training sample. The distribution of outcomes provides the model with the inductive bias about what a “small number” is.
>
> *Q2. For B=2021=43×47, is there a possibility that smaller primes are learned faster than 43,47,432,…?*
>
> For B=2021, the two divisors of the base, 43 and 47, are learned long before small primes are grokked. Over three different initializations of the model, 43 and 47 always get learned around epoch 20 and 22. 2 is learned at epochs 261, 442 and 125, and 3 at epochs 450, 182 and 226. Even with log-uniform outcomes, grokking always sets in after the divisors of the base are learned.

---

> > ### Author Response · Authors · 2023-11-19
> > **Reply to reviewer t8xf (2/2)**
> >
> > *Q1. It's said that composite bases allow one to check for divisibility by seeing the rightmost digits.*
> >
> > We can clarify the situation about composite bases. As explained above, if $B^n=kf$, the model can check divisibility by $f$ by comparing the $n$ last digits of a number with the $k$ values $0, f, 2f, (k-1)f$. For a given base $B$, divisibility will be learned so long $k$ (and $n$) are not too large (or the model cannot memorize the $k$ values).
> >
> > For B=210, divisibility by 4 and 9 are learned because they can be tested on the two last digits, from lists of 11025 and 4900 multiples. On the other hand, divisibility by 8 and 27, which require three digits, are not.
> >
> > For base 60, divisibility by 2 and 4 (tested on the last digit), and by 8 and 16 (tested on the 2 last digits) are learned. Divisibility by 32, which requires 3 digits, is not. (See also see our reply to Q3 below)
> >
> > For base 350, we expect neither 3, nor 8 to be learned in the initial phase.
> >
> > *Q3. Also given the list intuition, why 420 is performing better than 210 in Table 2?*
> >
> > This is because 420 is divisible by 4, and not 210. Table 19 in appendix E2 indicates that for B=420, 8 and 16 (which can be tested on the last two digits) are correctly predicted, but not 32 (which requires three digits). For B=210, only 2 and 4 (and their multiples) are correctly predicted.
> >
> > As a result, in B=420, models will correctly predict GCD that are multiples of 8 and 16 (and other divisors of the base, e.g. 24, 48, 40, 80…), but B=210 cannot. This explains the different in performance.
> >
> > *Q4. Further, the list intuition does not seem to be applicable anymore in Table 6 (E.g., 2401=7^4 is having the best performance). What could be the reason for this?*
> >
> > The list intuition is specific to divisors of the base. For 2401, 7 and 49 are learned this way, but all other divisors are learned by grokking. Performance therefore depends on how fast the model groks the small prime, a quantity that varies a lot with model initialisation (figure 5 in appendix E1 shows learning curves for 3 different model initializations, for B=2023).
> >
> > In other words, while it is clear that large base learn faster, randomness in model initialization is probably the main reason 2401 achieves better results than other large bases.
> >
> > *Q5. In claim U2, one can see that 21 is wrongly classified into C4 .*
> >
> > The 21 is a typo. After checking, it is 24. Thank you for spotting it. We did not find exceptions to the classification rules U1, U2 and U3.
> >
> > *Q6. In the line before table 5, we see that for B=30; B−1,B+1 are learned faster than other primes.*
> >
> > For B-1 and B+1, simple rules exist for all bases: the sum of the digits should be a small multiple of B-1 (because $B^k = 1 \mod B-1$), and the alternating difference $x_1-x_2+x_3-x_4+..$. should be a small multiple of B+1 (because $B^k = (-1)^k \mod B+1$). Similar rules exist for other divisors: e.g. the divisibility by 3 rule in base 10. However, all these rules involve all digits in the representation, and imply that the model must have learned to add or subtract.
> > This result for B=30, also observed in one experiment for base 31, suggests that such divisibility rules can indeed be learned, but that the usual “learn divisors of B, then grok small primes” is easier.
> >
> >
> > *Q7. What stopping criterion has been used?*
> >
> > We did not use a stopping criterion, and let the models run for a fixed number of epochs (mentioned in the results). We agree that longer training would achieve better results.
> >
> > *Q8. The addition and multiplication of rationals mentioned in the intro and studied in the appendix require the output to be simplified. So I wonder, is there any particular difference or additional complexity comparing to the simplification task itself?*
> >
> > Multiplication is more complex, in the sense that the two pairs of operands must be multiplied before the result is simplified.
> >
> > Addition is even more complex, because the operands must first be multiplied so that they have a common denominator, then numerators need to be added, and finally the result must be simplified.
> >
> > This said, our results on fraction comparison, which involve computing ad-bc when comparing a/b and c/d, suggest that transformers can handle addition and multiplication.
> >
> > *Q9. What is the positional embedding used for the model?*
> >
> > We use absolute encodings of positions, positional embeddings are learned during training.
> >
> > *R1. My understanding is that the samples at each batch are freshly generated. I think it would be beneficial if this is communicated more clearly and earlier to the reader.*
> >
> > We will clarify this
> >
> > *R2. Table 5 is a bit misleading with the current format.*
> >
> > The new divisors learned are “new” compared to the natural outcome distribution. In the B=1000 case, 3 was learned with the natural distribution of outcomes. We will clarify this.
> >
> > *R3. For Section 5 experiments in page 6, it would helpful if training loss plot is provided.*
> >
> > We will add these curves in the appendix

---

> > > ### Comment · Reviewer_t8xf · 2023-12-05
> > >
> > > I thank the authors for their response and the revised version of the paper. I agree with reviewers 21KR and zrzn that the intuitions provided in the paper seem to be limited to the GCD task. The paper reports how transformers perform on the GCD task but it does not explain why transformers learn in this particular way. Nonetheless, I think the experimental results (especially the dependency on different input and output distributions) can be insightful for people working on the mathematical abilities of transformers. As a result, I am increasing my score to borderline accept.

---

### Official Review · Reviewer_QHKu · 2023-11-06

**Soundness:** 4 excellent
**Presentation:** 4 excellent
**Contribution:** 4 excellent
**Rating:** 8
**Confidence:** 4

**Summary:**

The paper focuses on training Transformers from scratch to predict the GCD of two numbers and explaining the algorithm used by the trained model.  It is observed that for almost all pairs of numbers for which the GCD is k, the model outputs a unique value f(k).  The model learns a set of numbers D, and for an input pair with gcd k, f(k) is the largest number in D that divides k. When numbers are represented in base B, set D contains all the products of primes dividing B. For large bases B, if training continues for very long, set D starts to include other prime numbers as well, usually learned in a monotonically increasing fashion. For example, in base 1000, set D would have numbers 1, 2, 4, 5, 10, 16, 20 and so on. If the training is continued for long, this set also starts to include 3 and its multiples with the numbers already in D. If the GCD k is one of the elements of this set, then the model output will be correct, otherwise, the model outputs the largest element of this set that divides k. The paper also investigates the role of training distribution.

**Strengths:**

I really enjoyed reading this paper. In most cases, the algorithms learned by Transformers do not seem interpretable. I was surprised by how structured the encoded algorithm is in the case of GCD computation. This structure is very well explored in the paper through cleverly designed experiments and the intuition behind the results is also explained well.

**Weaknesses:**

I don't see any substantial weakness. Perhaps one could say that the implications of these results for large language models are unclear. However,  we barely know anything about the internal workings of Transformer models (the architecture behind LLMs) and this paper makes a small (and interesting!) step in enhancing our understanding.

**Questions:**

Minor suggestions/clarifications:

1. On page 2, the term epochs is used for training, which suggested to me that there is a fixed training set over which the model is trained for multiple epochs. But in the last paragraph of the paper, it is mentioned that new training data is generated on the fly. Perhaps this should be clarified on page 2.

2. In the abstract and introduction of the paper, there are lines of the form ``Models trained from uniform operands only learn a handful of GCD
(up to 38 out of 100)''. This is confusing if the reader does not know that all numbers with the same GCD are mapped to the same output by the model. Without this information, it can be the case that two pairs of numbers have the same GCD but the model output is correct for only one of them. In that case, it is unclear what would it mean for the model to learn a handful of GCDs. This should be made clear in the abstract and intro.

---

> ### Author Response · Authors · 2023-11-19
> **Reply to reviewer QHKu**
>
> Thanks you for your comments, and suggestions. We will integrate them in the revised version (updated in a few days).
>
> *Perhaps one could say that the implications of these results for large language models are unclear.*
>
> We agree that immediate applications to LLM are not obvious. We believe that the log-uniform sampling of operands and outcomes could be of help when fine tuning LLM on mathematical tasks (in fact, this might already be the case, since numbers scraped from the Internet tend to follow a Zipf/Benford law).
>
> A more likely application of this research is for Foundational models for Science, large language models pre-trained on a large corpus of scientific facts (equations, calculations, scientific data), to be used in Mathematics, Physics or Chemistry. These models need to perform correct integer calculations, and our results show how they can be learned.
>
> Finally, our main research direction is model explainability. Most current approaches focus on mechanical interpretability (i.e. understanding model weights). We propose a different direction, which focuses on elucidating model predictions for specific tasks. We believe mathematical problems are a good playground for such studies, because we can leverage our theoretical understanding of the underlying problems. Extending our approach to a larger set of mathematical problems might provide interesting insight about the algorithms learned by transformers (and other deep learning architectures).

---

> > ### Comment · Reviewer_QHKu · 2023-11-21
> >
> > I thank the authors for further clarifying the main contribution of the paper. I will take this into account while engaging with other reviewers and deciding the final score for the paper.

---

### Official Review · Reviewer_21KR · 2023-11-10

**Soundness:** 4 excellent
**Presentation:** 3 good
**Contribution:** 2 fair
**Rating:** 5
**Confidence:** 4

**Summary:**

This paper trains small transformer models (4 layers) to compute the GCD of a pair of numbers. This is framed as a sequence prediction task. The input is a pair of numbers (a, b) encoded as a sequence of numbers in a base B. The model predicts a sequence of numbers (in base B) which collectively represent the GCD of (a, b). Performance is measured by 2 metrics - accuracy in prediction GCD and number of integers between 1 to 100 which are a GCD prediction. The second metric is required according to the paper because for any given pair (a, b), the GCD is likely to be small and they also turn out to be harder to predict correctly. The authors perform a bunch of experiments in a variety of configurations which will be described below.

For the first set of experiments, the training dataset (pairs of numbers) is sampled uniformly between 1 to a million. Models trained on these datasets show pretty variable accuracy but the main trend seems to be that using larger bases of composite numbers to encode pairs work well. Models also exhibit the property of being able to predict GCDs equal to the (prime) divisor of their bases (and their small powers) pretty well. Other small prime numbers which are not divisors of the base are ‘grokked’ much later during training.

Next, the authors decide to oversample pairs of small numbers to create a lop-sided training distribution (called log-uniform in the paper). Models trained on this kind of distribution show dramatically improved and consistent performance regardless choice of base.
Finally, the authors modify the training distribution to also be uniform over the predicted GCD. This maintains good accuracy of these models but degrades the consistency with which model predictions could be explained.

To analyze the model predictions, the authors introduce ‘rules’ which they obtain after looking at the results after each stage of the above experiments. These rule help us explain model predictions in a uniform fashion.

**Strengths:**

1. The main idea is pretty straightforward and most of the concepts in the paper are well explained (see concerns discussed below).
2. To my knowledge, computing and analyzing GCD prediction via transformers has not been done before.
3. Very comprehensive suite of experiments and exhaustive analysis. I appreciate the authors performing such a wide range of experiments. I also really like the nice link between theoretical accuracy and practical accuracy provided in Appendix C.
4. Claims made by the authors are clearly backed up the experiments in the paper (see some minor concerns below).

**Weaknesses:**

My main concern about this work lies with the significance of the results and observations made. The authors train a transformer to predict the GCD which seems to work fairly well with some tricks in picking the right dataset. However, I’m not convinced about why this result would be of significant interest to the wider community and what it says about the representational power of transformer themselves beyond the narrow context of learning GCDs. Like I mentioned earlier, the experiments are thorough and the analysis is extensive but I’m struggling to understand the value of this beyond this specific task.

The authors present ‘3 rules’ which supposedly explain model predictions and while they seem technically correct, they do seem pretty specific to the task and don’t seem to point to something specific about transformers themselves. For example, do we get these same rules if we swap the transformer architecture to something else? Is there something special about the architecture?

Having said all of that, I appreciate a good, rigorous experimental paper and I’m open to being convinced by the authors and reading other reviewers comments about the value of this work.

Other:
1. The authors hypothesize on Page 3 “For prime bases, such as B = 2, the three rules suggest that the model learns to predicts GCD by counting the rightmost zeros in its inputs”. Something like this seems like it should be fairly easy to confirm by visualizing the attention weights. I wonder if the authors have tried this.
2. The authors mention that their model is ‘fully explainable’ (see abstract). I was expecting the model to output some kind of explanation for its prediction (say via attention) but what the authors actually meant was that the model predictions can be $\textit{explained by}$ a human  because they follow certain rules. These rules keep on changing depending on the training configuration and can only be inferred after a thorough analysis of the results so I’m not sure if we can call it explainability at least in the traditional sense used in XAI. However, the authors do say in the final section “Our approach to explainability differs from most works on the subject. Instead of looking at model
parameters, we engineer experiments that reveal the algorithms that the model is implementing”. I think a statement of this sort at the start of the paper would help in clarifying the confusion. Regardless, I’m not sure if I would call a model showing some consistent trends in the output as ‘explainable’.

Minor:
1. On page 2, when introducing the procedure to generate the stratified test set, the symbol ‘M’ is introduced without any explanation as follows “Sample a and b, uniformly between 1 and M/k , such”. I’m assuming M is the upper limit on the numbers a and b. This symbol is used throughout the paper and explained nowhere so it’s confusing.
2. The authors use two metrics to evaluate their models - accuracy on a test set and number of correctly predicted GCD below 100. At several points in the paper, terms like “50 GCD” or “50 correct GCD” are used. I now understand that they mean that the model predicts 50 GCDs out of a 100 between 1 to 100 correctly but it was definitely confusing to me initially. I think clarifying these terms at the start can be useful.
3. Typo - Appendix C, first line, “for models from section 3 the follow the three rules” should be “for models from section 3 that follow the three rules”

**Questions:**

How is the choice of the base made? Are they also sampled from a distribution?

---

> ### Author Response · Authors · 2023-11-19
> **Reply to reviewer 21KR**
>
> Thank you for your comments, questions and suggestions. We will integrate our replies in the revised version (in a few days).
>
> *My main concern about this work lies with the significance of the results and observations made.*
>
> The broader context of our research can be summarized as follows:
>
> * This is a first step towards using mathematical tasks to understand deep learning models, by investigating their predictions on selected tasks. We see this as a new approach to explainability, and demonstrate its feasibility on a non-trivial mathematical task.
> * While mathematical tasks are not the main focus of LLM practitioners, they play a central role for the development of Foundational Models for Science: large language models pre-trained on mathematical equations. Our results show how transformers can be trained to perform exact calculations involving integer divisibility, a central task in integer arithmetic and number theory.
> * Our results on training distributions, i.e. the fact that some distributions allow for faster learning and better out-of-distribution performance, may apply to other arithmetic tasks. In particular, log-uniform operands and outcomes could be used when fine tuning LLM, or training foundational models for physics.
>
>
> *Are the 3 rules task specific? Do they apply to other architectures?*
>
> We agree that the 3 rules are specific to tasks involving integer divisibility. Still, they provide insight about what our models can learn. In section 3, the model only learns divisors of the base, that can be tested by looking at a small number of digits. Divisibility by 3 or 9 (in base 10), which can be tested by summing all digits in the representation, seem harder to learn. For a transformer, we believe this suggests that the attention mechanism helps learn properties that concentrate on a few tokens in the input sequence, but struggle with  non-local properties, such as the sum of all digits in an operand.
>
> Thank you for suggesting additional experiments on architectures! We experimented with LSTM and GRU, with 1024 and 2048 dimensions, and 4 layers, for 10 different bases (from 10 to 2401). We will include our findings in the appendix of the revised paper, but initial results suggest that our observations extend to these recurrent networks.
>
> When training on uniform operands for 260 epochs, we observe that:
> * for B=420, 34 GCD below 100 are learned, products of {1,2,4,8,16},{1,3,9}, {1,5} and {1,7} (divisors of the base)
> * for B=210, 31 GCD are learned: products of {1,2,4,8}, {1,3,9}, {1,5}, {1,7}
> * for larger bases, grokking seems to happen as well: for B=2021, 2 and 3 are grokked, for B=2023 and 2025, 2 is grokked, for B=2401, 2 and 4 are grokked.
> When training on log-uniform operands, 37 GCD are learned after 150 epochs, with uniform outcomes 57 GCD under 100 are learned.
>
> Overall, this suggests that our results extend beyond the transformer architecture.
>
> *Whether the model counts rightmost zeroes should be fairly easy to confirm by visualizing the attention weights. I wonder if the authors have tried this.*
>
> We did, and observed that some attention heads do focus on the low digits of the operands. We hesitate to jump to conclusions because:
> * input sequences are short, and different heads attend to almost all tokens in the sequence,
> * this pattern was observed in tasks that do not involve divisibility,
> * the same phenomenon is observed in architectures without attention (see above).
>
> *Regardless, I’m not sure if I would call a model showing some consistent trends in the output as ‘explainable’.*
>
> We believe ours is a valid alternative to mechanical interpretability. Even though the low level specifics of the model are not fully understood, the 3 rules account for more than 99.5% of model predictions. We believe this qualifies as a valid explanation. Such a situation is common in physical sciences. Many macroscopic phenomena, the relation between volume, pressure and temperature in a gas, or the transformation of an egg into a fully developed individual, can be explained, even though the low level processes, the trajectories of particles in the gas or the biochemical processes of embryogenesis, are not completely elucidated. We see our approach as a new direction in explainability.
>
> *How is the choice of the base made? Are they also sampled from a distribution?*
>
> In section 3, we selected small bases (2,3,4,5,6,7), bases often used in practice (10, 100, 1000), bases with many small prime divisors (30, 60, 210, 420) and primes bases close to these values (31, 211, 997).
>
> For section 4, we selected bases depending on their prime decomposition. In particular, we focused on bases not divisible by some small primes, in order to observe grokking. The first column of table 4 provides an explanation of sorts.
>
> In section 5 and 6, we reused the bases from the previous sections, for the sake of comparability.

---

> > ### Comment · Reviewer_21KR · 2023-11-23
> > **Response to rebuttal**
> >
> > I thank the authors for their response.
> >
> > However, while I appreciate the results in the paper, I am still not convinced by the significance of the results beyond the GCD context. I will maintain the score for now.

---

### Author Response · Authors · 2023-11-21
**Revised version**

We thank again the reviewers for their constructive comments. We attached a revised version of the paper, which addresses most of their comments and our replies.

Main changes are :

* in the introduction, the contributions were clarified, so as to better explain when explainability holds and fails, and the fact that the model learns to predict a unique value for all pairs with the same GCD.
* in the introduction, we added a section on the broader motivation and potential impact of this research
* in the base experiments, we described the shortcuts used by the model for composite bases (instead of focusing on prime bases only)
* in the discussion, we rewrote the sections on explainability and the description of the algorithm the models implement
* in the appendix (E.2) we added loss and learning curves for models trained on log-uniform operands, and log-uniform operands and outcomes (for B=2023, for the sake of comparability with the curves in appendix (E.1)
* we added in appendix D.5 the results of our experiments with LSTM and GRU.

---

### Meta-Review · Area_Chair_sYvH · 2023-12-06

**Metareview:**

This work studies Transformers trained from scratch on the synthetic mathematical task of computing GCD. The authors aim to understand the functional behavior of the trained model — that is, the input-output map that is actually learnt during and at the end of training. They manage to characterize the learnt function fairly well, by comparing it to a “sieve-like” algorithmic for GCD. This allows them to predict the failure inputs of the model, and the evolution of these failures throughout training.

Reviewers agreed on the novelty and strength of this work (and I concur): Understanding the failure modes of models even on simple tasks is a notoriously difficult problem, so it is surprising that such a strong characterization was found (even for the simple case of GCD). Reviewers also appreciated the experimental rigor of the work, and had no concerns about technical correctness.

The main concern of reviewers was in the significance of this work: the problem setup is very simple, and it is unclear what this analysis teaches us about other (perhaps more realistic) tasks.
However, in this case, I believe the strengths outweigh this weakness. Our understanding of Transformer learning is in its infancy, and so it can be useful to start with simple, fully controlled experimental settings where we can gain complete understanding. Such observations, while small in scope individually, serve as useful datapoints which may eventually weave into a more general understanding of learning.

This work is also notable in that it studies only the external input-output behavior of Transformers, avoiding any heuristic discussion of Transformer internals (attention maps/etc). This approach differs from many popular approaches in mechanistic interpretability. The paper thus serves as a good example of how structure can be identified from careful black-box experimentation, without involving interpretation of internals (which are often subjective).

Thus I recommend acceptance. In the camera-ready, I encourage the authors to incorporate all feedback from reviewers. I strongly suggest changing the word "explainable", since this has other connotations which could confuse readers (consider e.g. "we characterize the function learnt...").

**Justification For Why Not Higher Score:**

The scope of this work is limited to a very simple task, and the broader implications are currently unclear.

**Justification For Why Not Lower Score:**

The results are very strong in comparison to most other work on mechanistic interpretability / understanding the operation of LMs (even on simple tasks).

---

### Decision · Program_Chairs · 2024-01-16

Accept (spotlight)